# Biosynthesis, Quantification and Genetic Diseases of the Smallest Signaling Thiol Metabolite: Hydrogen Sulfide

**DOI:** 10.3390/antiox10071065

**Published:** 2021-07-01

**Authors:** Joanna Myszkowska, Ilia Derevenkov, Sergei V. Makarov, Ute Spiekerkoetter, Luciana Hannibal

**Affiliations:** 1Laboratory of Clinical Biochemistry and Metabolism, Department of General Pediatrics, Adolescent Medicine and Neonatology, Medical Center, Faculty of Medicine, University of Freiburg, 79106 Freiburg, Germany; joanna.myszkowska@uniklinik-freiburg.de; 2Department of Food Chemistry, Ivanovo State University of Chemistry and Technology, 153000 Ivanovo, Russia; derevenkov@gmail.com (I.D.); makarov@isuct.ru (S.V.M.); 3Department of General Pediatrics, Adolescent Medicine and Neonatology, Medical Center, Faculty of Medicine, University of Freiburg, 79106 Freiburg, Germany; ute.spiekerkoetter@uniklinik-freiburg.de

**Keywords:** hydrogen sulfide, transsulfuration, sulfur metabolite, genetic disease, cystathionine beta-synthase, metabolism

## Abstract

Hydrogen sulfide (H_2_S) is a gasotransmitter and the smallest signaling thiol metabolite with important roles in human health. The turnover of H_2_S in humans is mainly governed by enzymes of sulfur amino acid metabolism and also by the microbiome. As is the case with other small signaling molecules, disease-promoting effects of H_2_S largely depend on its concentration and compartmentalization. Genetic defects that impair the biogenesis and catabolism of H_2_S have been described; however, a gap in knowledge remains concerning physiological steady-state concentrations of H_2_S and their direct clinical implications. The small size and considerable reactivity of H_2_S renders its quantification in biological samples an experimental challenge. A compilation of methods currently employed to quantify H_2_S in biological specimens is provided in this review. Substantial discrepancy exists in the concentrations of H_2_S determined by different techniques. Available methodologies permit end-point measurement of H_2_S concentration, yet no definitive protocol exists for the continuous, real-time measurement of H_2_S produced by its enzymatic sources. We present a summary of available animal models, monogenic diseases that impair H_2_S metabolism in humans including structure-function relationships of pathogenic mutations, and discuss possible approaches to overcome current limitations of study.

## 1. Introduction

### 1.1. Biological Chemistry of H_2_S

Hydrogen sulfide (H_2_S) is the smallest biological thiol. In the biosphere, H_2_S is both an environmental toxin and a signaling metabolite with increasingly investigated roles in human physiology. An excellent review about the timeline of H_2_S investigation from its discovery by Swedish-German chemist Carl Wilhelm Scheele until present is available from Csaba Szabo [1]. This review focuses on the biosynthesis and quantification of H_2_S and on possible disturbances caused by genetic defects of metabolism that impair its turnover in humans. Thiols are nucleophiles of generally good reducing capacity that can engage in one- and two-electron transfer reactions [2]. Reaction mechanisms involving the transfer of one electron produce the corresponding reactive thiyl radical species. Two-electron oxidation of thiols yields disulfide species. The reactivity of H_2_S is comparable to that of other low molecular weight thiols, including its undergoing reversible as well as irreversible modifications. Reactions of H_2_S with other reactive species, including oxygen- and nitrogen-centered radicals have been comprehensively reviewed elsewhere [3,4,5].

Hydrogen sulfide is a weak acid with pK_a_~7.0 (Figure 1). Recent Raman studies of sodium sulfide dissolved in hyper-concentrated NaOH (aq) and CsOH (aq) cast serious doubt on the widely assumed existence of S^2−^ (aq) [6]. Under physiological pH, H_2_S exists predominantly as a hydrosulfide anion (HS^−^). It has been estimated that approximately 70–80% of hydrogen sulfide exists as HS^−^ in plasma [7,8]. In the cytosol (pH 6.9–7.0), approximately 50% of hydrogen sulfide exists as HS^−^. In specific organelles, the proportion of the different ionized forms of hydrogen sulfide may vary. For example, in lysosomes (pH 4.7–5.0), H_2_S would be the predominant species, whereas in the more alkaline microenvironment of the mitochondrial matrix (pH > 7.8), HS^−^ has been estimated to represent 92% of all available sulfide.

In this review, the term H_2_S refers to the sum of H_2_S/HS^−^. The signaling properties of H_2_S have often been compared to those of the gasotransmitters nitric oxide (NO) and carbon monoxide (CO) [9]. Among the similarities, all three signaling molecules are gases, are synthesized by enzymes and possess in vivo half-lives of seconds to minutes. The major difference that prevails until present time is the absence of a conserved canonical receptor molecule for H_2_S, which is in contrast to the case of NO and CO, both of which exert their functions via the sGC/cGMP signaling axis [9]. An excellent review on H_2_S signaling in the context of other known gasotransmitters is available elsewhere [10,11,12].

### 1.2. Concentration, Signaling and Cellular Response to H_2_S

As a redox signaling metabolite, the effects of H_2_S in physiology are determined by three major factors, namely, concentration, compartmentalization and reactivity [13]. *Firstly*, the concentrations of H_2_S in cells and extracellular compartments are maintained in a narrow physiological range by modulation of its biogenesis (transsulfuration pathway and microbiome production) and catabolism (mitochondrial oxidation) (Table 1). The occurrence of mutations in genes encoding enzymes responsible for the biosynthesis and catabolism of H_2_S can increase and reduce the concentration of H_2_S outside optimal homeostatic concentration. The exact contribution of the microbiome to overall H_2_S concentration is a matter of active investigation [14,15,16]. It is noteworthy that the reported concentrations of H_2_S in biological compartments vary substantially across laboratories. It has been noted that the reported concentrations of H_2_S have decreased over time [2], likely due to the optimization of methods for its quantification (see also Table 1). However, these technical barriers are far from overcome. Publications reporting plasma concentrations of H_2_S in the middle to high micromolar range continue to emerge [17,18,19,20], which complicates data interpretation and comparisons between different laboratories. Plasma concentrations of H_2_S > 10 µM are an overestimation that hampers meaningful interpretation of H_2_S effects in physiology. The same applies for cerebrospinal fluid, for which concentrations of ca 8.7 µM and 102 µM have been reported in pediatric and adult human subjects, respectively [21,22]. Excellent reviews on what may constitute physiological concentrations of H_2_S are available [23,24]. Details on existing methods of quantification of H_2_S have been covered in prior reviews [12,23,25,26,27] and updates are provided in the following sections of this review. *Secondly*, the physiological actions of H_2_S are dictated by its diffusion capacity away from the respective sites of biosynthesis, or in other words, its compartmentalization. The solubility and diffusion of H_2_S across biological membranes has been studied and reviewed comprehensively (Figure 2) [28,29]. One study utilized a 3D spherical diffusion model representing “a mammalian cell” that produces H_2_S in a continuous manner from its surface. By applying a condition that involves free diffusion of H_2_S in aqueous medium, without membranes, the authors determined that H_2_S would reach 260–590 neighboring cells. Introduction of resistance by lipid membranes at pH 7.4 was shown to predict a slowdown of diffusion in tissues (based on this 3D model of the cell), leading to local accumulation of H_2_S near the source [28]. The authors estimated that the ionization states of H_2_S modify the apparent diffusion coefficient from 1.51 × 10^−5^ cm^2^/s at pH 7.4 to 1.94 × 10^−5^ cm^2^/s at pH 6.5. Considering the existence of membranes and pH 7.4, H_2_S has a predicted diffusional capacity that extends up to 200 cells away from a single producing cell over a biosynthesis time span of 1 s (with a membrane permeability coefficient of 3 cm/s). This property supports the classification of H_2_S as a paracrine signaling molecule [28]. An independent mathematical model of H_2_S metabolism suggested that H_2_S is consumed mainly by mitochondrial oxidation and by diffusion out of the cell, with the former process being the primary regulator of H_2_S concentration [30]. Within the constraints of necessary mathematical assumptions (such as “working” concentrations of H_2_S, single exponential kinetic course, pH changes, approximate H_2_S diffusion distances), it was postulated that very little H_2_S diffuses out of cells, which would be mirrored by little if any H_2_S in blood [30]. In vitro studies indicated that both H_2_S (simple diffusion) and HS^−^ (facilitated diffusion by anion exchange protein AE1 in erythrocytes) produced by tissues enter red blood cells rapidly, thereby acting as a sink for H_2_S and lowering extracellular concentration of the gasotransmitter [31]. As can be appreciated from these in-depth experimental studies and predictions, the properties and relationships of the sources and sinks of H_2_S are complex. The extent and effects of H_2_S mobilization in the biological milieu require further investigation. 

The physiological effects of varying H_2_S concentrations appear to be specific for cell type and the dose (Table 2). Elevation of H_2_S appears to be associated with pro-inflammatory effects [14,43,44], whereas low to normal concentrations of H_2_S exert anti-inflammatory effects [45,46]. Likewise, the pharmacological actions of H_2_S follow a bell-shaped curve (reviewed in [47]). Both the inhibition of endogenous H_2_S biosynthesis (that has stimulatory effects on ATP production, being cytoprotective) as well as supplementation with exogenous pharmacological donors of H_2_S (that enhances cytostatic and cytotoxic actions) afford anti-cancer effects [47].

*Thirdly*, H_2_S and a series of reactive sulfur species (RSS), including polysulfides, persulfides and thiosulfates, can modify target proteins and induce specific physiological effects [59,60]. The persulfidation reaction appears to be the major pathway for sulfur-mediated post-translational modification of proteins [11]. For example, persulfidation stabilizes superoxide dismutase (SOD) against oxidation-induced aggregation without affecting its catalytic activity [61]. Persulfide modifications on proteins are reversible and, unless sequestered, labile. Interchange reactions with glutathione (GSH), thioredoxin (Trx) or cysteine can undo persulfidation [11]. In mammals, sulfurtransferases involved in sulfide synthesis and oxidation produce active site persulfides. On the other hand, polysulfides, which are more reactive than H_2_S, activate transient receptor potential (TRP) A1 channels, inhibit lipid phosphatase PTEN52 and activate transcription factor Nrf2 by modification of Keap1 [62]. The specificity of this mechanism demands further investigation. In bacteria, polysulfides and elemental sulfane sulfur can be produced during sulfide oxidation by *sulfide quinone oxidoreductase* (*SQOR*) [63]. In contrast, mammalian *SQR*s form persulfide rather than polysulfide [64,65,66]. The modification of the target protein takes place either directly or indirectly via a secondary carrier.

Besides RSS with S-S bonds, hydrogen sulfide can form other reactive species that contain sulfur-oxygen bonds [67,68,69]. Key intermediates in the oxidation of sulfide to sulfate are the so-called small oxoacids of sulfur (SOS)—sulfenic HSOH and sulfoxylic S(OH)_2_ acids. These SOS appear relatively long lived in aqueous solution, and thus may be involved in the observed physiological effects of H_2_S.

## 2. Biogenesis, Catabolism and Reservoir of Hydrogen Sulfide in Humans

### 2.1. Enzymatic Production of Hydrogen Sulfide

Biogenesis of hydrogen sulfide in humans occurs via two major routes: by endogenous specialized enzymes and as an end-product or intermediate of microbial metabolic pathways within the gut microbiota [10].

Endogenous production of hydrogen sulfide in mammals is carried out by three enzymes: *cystathionine*
*β-synthase* (CBS), *cystathionine*
*γ-lyase* (CSE, also known as cystathionase) and *3-mercaptopyruvate sulfurtransferase* (MST) [10,70]. Enzymes CBS and CSE are localized in the cytosol whereas MST resides in the mitochondrion (Figure 3). 

CBS and CSE are pyridoxal 5′-phosphate (PLP-) dependent enzymes of the transsulfuration pathway that convert homocysteine into cysteine. All available homocysteine in the cell derives from methionine catabolism [71].

CBS catalyzes the β-replacement of serine (Ser) by homocysteine to form cystathionine. Downstream, CSE catalyzes the conversion of cystathionine to cysteine, α-ketobutyrate and ammonia in a α, γ-elimination reaction. In addition, CBS and CSE participate in various side reactions, which generate the signaling molecule hydrogen sulfide (H_2_S) as a side product (Figure 4) [72]. Formation of cystathionine from L-Cys and L-Hcy yields H_2_S both in the reactions catalyzed by CBS and CSE. In addition, CSE catalyzes the formation of the products lanthionine and homolanthionine from substrate combinations Cys/Cys and Hcy/Hcy, respectively, both of which produce H_2_S. Thioether metabolites lanthionine and homolanthionine have been proposed as biomarkers of hydrogen sulfide generation in patients with CBS deficiency [73].

The third PLP-dependent pathway involves the enzyme *aspartate aminotransferase* (AAT), which has *cysteine aminotransferase* (CAT) activity [74]. In combination with the PLP-independent enzyme *3-mercaptopyruvate sulfurtransferase* (MST), sulfane sulfur can be generated, which, after reduction, can liberate hydrogen sulfide.

#### 2.1.1. CBS

*Cystathionine**β-Synthase (*CBS*)* is the first enzyme of the transsulfuration pathway in which Hcy is converted to Cys [75]. A deficiency of this enzyme due to an inherited mutation leads to accumulation of Hcy to toxic concentrations. Deficiency of CBS is the most common cause of homocystinuria, an inherited metabolic disease with an impact on almost all human organs [75].

*CBS* is mainly expressed in the brain, liver, kidney and pancreas (Figure 5). It is predominantly localized in the cytosol, but in specific cell lines CBS could be also found in the nucleus [76] and the mitochondrion [77]. This enzyme produces from 25 to 70% of the total concentration of endogenous H_2_S [78].

The human *CBS* gene is located on chromosome 21 (i.e., 21q22.3) and it encodes a protein of 551 amino acids. The active form consists of four 63-kDa subunits [79,80]. Each subunit contains three structural domains, namely, N-terminal domain harboring heme, the catalytic domain containing PLP and the C-terminal domain with regulatory functions. The N-terminal domain binds to the heme-cofactor and enables successful protein folding and assembly, but it is not necessary for the catalytic activity of the enzyme [81]. The heme in CBS occurs in two oxidation states: ferric (Fe^3+^) and ferrous (Fe^2+^). Ferrous CBS can bind the gas signaling molecules, CO and NO, which results in inhibition of its catalytic activity [82]. A recovery of the ferric heme state and its enzymatic activity is possible by air oxidation of the ferrous-CO or ferrous-NO forms of CBS. The catalytic domain of the enzyme has a binding site for another cofactor, PLP [75,79]. The C-terminal domain is the regulatory part and it consists of two CBS motifs (CBS1 and CBS2) that dimerize to form a Bateman domain. It is responsible for CBS subunit tetramerization and it contains the binding sites for the allosteric activator S-adenosylmethionine (SAM) [79,83]. SAM is a universal methyl group donor that supports over 200 methyltransferase reactions [84,85]. Allosteric modulator SAM stabilizes CBS [86] and can increase its enzymatic activity up to 5-fold [87]. In addition, SAM was shown to modulate binding of gasotransmitters NO and CO to the heme of CBS as another means to regulate enzymatic activity [88]. Another form of CBS regulation is post-translational modification by a small ubiquitin-like modifier (SUMO) protein, which is correlated with the localization of CBS in the nucleus [76,89]. Sumoylation of CBS depends on substrate concentration and leads to reduced catalytic activity [70]. All in all, the activity of CBS seems to be highly regulated. Still, H_2_S production by CBS appears to be independent of blood Hcy concentrations, possibly related to the fact that the affinity of CBS for Hcy is lower than its affinity for serine or cysteine [10]. 

In terms of substrate specificity, CBS participates in the first and committing step of the transsulfuration pathway: the β-replacement of serine with homocysteine, yielding cystathionine and water [78]. Instead of serine, cysteine can also act as a substrate, which results in the formation of cystathionine and H_2_S as reaction products. In addition, CBS catalyzes side reactions that produce H_2_S from cysteine: the β-replacement of cysteine by water to form Ser and the β-replacement of Cys by a second mol of Cys to produce lanthionine. In both of these reactions, H_2_S is eliminated and SAM enhances β-replacement by Hcy of Ser and Cys. Out of these reactions, the canonical β-replacement reaction of Cys with Hcy is kinetically most efficient toward the production of H_2_S [78] (Figure 4).

#### 2.1.2. CSE

*Cystathionine**γ-lyase (*CSE) is the second enzyme of the transsulfuration pathway, downstream from CBS. The main reaction of CSE is to cleave cystathionine to form cysteine, ammonia and α-ketobutyrate. Like CBS, CSE catalyzes side reactions that lead to the formation of H_2_S, such as β-elimination of Cys, which is estimated to be the major reaction driving H_2_S synthesis by CSE [3,78] (Figure 4).

*CSE* is mainly expressed in the liver, kidney, thoracic aorta, ileum, portal vein, uterus, brain, pancreatic islets and the placenta (Figure 5) [90,91,92,93,94]. Based on recent studies, CSE is the primary source of H_2_S in the peripheral vasculature [95] and it shows modest presence and activity in the brain [94]. 

Human CSE is a tetramer with a subunit molecular mass of 45 kDa [89]. Each subunit contains two CXXC motifs, whose influence on CSE activity via its predicted redox-sensing properties awaits experimental confirmation. Under in vitro conditions, CSE is post-translationally modified by SUMO [89], a signal that is often used for nuclear localization. The physiological relevance and the possible impact, if any, of sumoylation on CSE function has not yet been described. Calmodulin increases the activity of CSE by 2-fold, even in the presence of high blood calcium concentration (2 mM) [70,96]. The regulation of CSE is still not very well understood.

CSE belongs to the γ-family of PLP-dependent enzymes and it catalyzes, as mentioned before, α,γ-elimination of cystathionine to give Cys, α-ketobutyrate and ammonia. A variety of CSE-catalyzed side reactions lead to H_2_S formation, including cysteine-dependent β-reactions and homocysteine-dependent β-reactions [97]. An alternative route for H_2_S synthesis from cysteine catalyzed by CSE has been proposed to involve β-elimination of cystine, which leads to the formation of thiocysteine as an intermediate product. Thiocysteine decomposes to H_2_S thereafter, via non-enzymatic reaction with other thiols [97]. Furthermore, studies of Chiku et al. identified two novel sulfur metabolites generated as byproducts of H_2_S synthesis by CSE: lanthionine and homolanthionine (Figure 4) [97].

Under normal conditions, approximately 70% of H_2_S is produced from Cys and the remaining 30% is from Hcy [97]. However, under conditions with higher concentrations of homocysteine, Hcy rather than Cys becomes the preferred main source for H_2_S synthesis. According to Yang & He, CBS^+/−^ mice showed upregulated CSE in the heart, suggesting that CBS deficiency induces CSE [98]. 

In addition, CSE has relaxed substrate specificity, which allows accommodation of cystathionine, cysteine and homocysteine in the same binding pocket where they compete for forming a Schiff base with PLP [78]. In contrast, the active site of CBS cannot accommodate two molecules of Hcy, but only one [78]. This has been proposed as the molecular basis by which H_2_S production by CSE but not CBS is sensitive to homocysteine concentrations [78]. The concentration of H_2_S generated by homocysteine-dependent reactions increases proportionally with the degree of hyperhomocysteinemia [97]. This implies that CSE may have a significant role in Hcy clearance at pathophysiological elevated Hcy concentration, particularly in tissues or organs with CBS deficiency [10]. Thus, under normal conditions (5–15 µM Hcy in plasma), CSE is responsible for ~30% of the H_2_S production in the transsulfuration pathway [78]. This amount can increase up to ~45 and ~74% under moderate and severe hyperhomocysteinemic conditions [78]. 

In contrast to CBS, the activity of CSE is highly dependent on the cellular supply of PLP (vitamin B6) [72]. Therefore, H_2_S production by CSE is limited under conditions of vitamin B6 shortage. In contrast, CBS is very resilient, maintaining much of its activity in all but severe states of vitamin B6 deficiency [72].

#### 2.1.3. MST

*3.-Mercaptopyruvate sulfur-transferase* (*3*MST) is the third main H_2_S-generating enzyme, which participates in cysteine metabolism [99]). It acts in conjunction with *cysteine aminotransferase (CAT*), which is identical to *aspartate aminotransferase (AAT*), to produce H_2_S. MST produces H_2_S from 3-mercaptopyruvate (3-MP), which is generated by *CAT* from Cys in the presence of α-ketoglutarate [98,99] (Figure 4).

*3. MST* is expressed mainly in the liver, kidney, heart, lung, thymus, testis, thoracic aorta and the brain (Figure 5) [100,101]. Western blot analysis and enzymatic assays revealed the presence of active MST, but undetectable CBS and CSE, in red blood cells [10,102] and the vascular endothelium [101]. In general, MST is described as a mitochondrial enzyme. Studies of Fräsdorf et al. described two cytosolic splice variants of the protein, MST-Iso1 and MST-Iso2, with the latter also being identified in the mitochondria of HEK293a and HeLa cells [10,103]. Neuronal cells expressing *3*MST and CAT exhibit resistance against oxidative glutamate toxicity [104]). The study suggested that H_2_S produced by *3*MST/CAT clears reactive oxygen species in mitochondria and preserves the function of cells from oxidative damage. 

In contrast to the enzymes of the transsulfuration pathway CBS and CSE, MST is inhibited under oxidative stress conditions [70]. MST has an active site cysteine residue, which is sensitive to oxidants.

Aspartate inhibits H_2_S formation by *3*MST/CAT [105]. This amino acid is a preferred substrate for CAT or AAT, thereby competing with 3 MST.

Recent work shows that thioredoxin could modulate MST activity as well [106]. Under physiologically relevant concentrations, reduced thioredoxin binds to MST and decreases its reaction rate constant for 3-MP compared to other sulfur acceptors. In contrast, under oxidizing conditions, oxidized thioredoxin increases the reaction rate and MST is free to interact with its substrate 3-MP. This could explain the observed stimulation of MST under oxidative stress conditions wherein the reduced form of thioredoxin is decreased [106].

The fraction of cysteine not employed for transsulfuration reactions, glutathione or protein biosynthesis, undergoes catabolism to yield cysteine sulfinic acid via oxidation of the thiol moiety by the enzyme cysteine dioxygenase [108]. Cysteine sulfinic acid is further processed via two distinct routes to produce taurine or sulfate as the final products, respectively. These pathways have been reviewed in detail elsewhere [108]. 

### 2.2. Non-Canonical Biogenesis of H_2_S

Besides the enzymes of sulfur amino acid metabolism, erythrocytes produce H_2_S from inorganic and organic polysulfides [109,110]. It has been estimated that erythrocytes can produce H_2_S at a constant concentration of 170 µmol (L cells)^−1^ min^−1^ [110]. Experimental formation of H_2_S was observed after addition of either GSH or nicotinamide adenine dinucleotide (NADH) or nicotinamide adenine dinucleotide phosphate (NADPH) to the cell lysate, which suggests a physiological role of H_2_S as an electron carrier and an evolutionary link between the eukaryotic cytoplasm and sulfur-reducing *Archaea* [110]. In addition, human erythrocytes convert garlic-derived organic polysulfides into H_2_S via glucose-supported and thiol- and glutathione-dependent reactions [109,111].

Studies by Akaike et al. demonstrated that mitochondrial enzyme *cysteinyl tRNA synthetase* (*CARS2*) plays a major role in catalyzing the conversion of Cys into Cys- per/polysulfide species (Figure 6) [112]. Cys-per/polysulfides are efficiently reduced by the thioredoxin and GSH enzyme machineries using NADPH as a source of electrons [113]. Therefore, these non-canonical enzymatic pathways could contribute significant H_2_S production in cells. 

Besides the systems described above, non-enzymatic formation of H_2_S was also demonstrated in blood, with specificity toward precursor metabolite cysteine [114]. The study reported cysteine-dependent formation of H_2_S via a reaction that requires pyridoxal phosphate and iron, including heme iron. The reaction progresses via formation of a cysteine-aldimine intermediate, and produces pyruvate, NH_3_ and H_2_S as the final products [114]. Results from the study suggest that this reaction contributes very little to systemic, basal H_2_S concentrations, but could prove detrimental in pathological conditions characterized by elevation of released iron such as is seen in hemolytic and hemorrhagic disorders [114]. In vitro studies indicate that met-hemoglobin scavenges and regulates sulfide pools in blood, including the maintenance of hemoglobin itself in its reduced, oxygen-binding form that is necessary for erythrocyte function [115]. The roles of sulfide pools in erythrocytes have been covered in detail elsewhere [116]. 

### 2.3. Hydrogen Sulfide Derived from the Microbiome

Besides enzymatic formation of H_2_S by cells, the microbiome in the human gut plays an important role in the biogenesis of hydrogen sulfide [14]. 

The intestinal microbiome metabolizes cysteine and methionine in order to produce sulfur-containing structures, metabolites, signal molecules and cellular energetics [14]. Sulfate-reducing bacteria utilize thiosulfate to generate H_2_S. In the colonic lumen, several bacterial species are able to convert Cys into H_2_S [117]. In fact, numerous bacterial groups (*Fusobacterium, Clostridium, Escherichia, Salmonella, Klebsiella, Streptococcus, Desulfovibrio,* and *Enterobacter*) transform Cys to H_2_S, ammonia and pyruvate by the enzyme *cysteine desulfhydrase* [16,118,119]. In particular, *Escherichia coli* and *Salmonella typhimurium* were described to have a lack in enzymes that convert methionine to cysteine. Therefore, those species are not able to use methionine as a source for growth [120]. In contrast, *Fusobacterium* has at least four genes which encode for different enzymes catalyzing the production of H_2_S [121]. One of the encoded enzymes is CSE which produces hydrogen sulfide and Ser from Cys. Another enzyme has been described as having *“L-cysteine lyase”* activity, perhaps similar to CSE. L-Cys lyase combines two cysteine residues to form lanthionine and H_2_S. Two distinct proteins with *cysteine desulfhydrase* activity have also been described [121]. 

Notably, a comparison of germ-free versus conventional mice showed that presence of the microbiota was associated with higher concentration of free H_2_S not only in some intestinal tracts (colon and cecum) but also in plasma [122]. Moreover, the conventional mice showed higher concentrations of bound sulfane sulfur in plasma, fat and lung, a higher CSE activity, and in contrast, lower cysteine concentration in most organs and tissues [122]. Only the aorta tissue in germ-free mice had a significant clear decrease in cysteine concentration. The data of Shen et al. suggested that the presence of the microflora had an impact on CSE activity and Cys bioavailability [122]. One possible explanation could be that bacterial products leak into the bloodstream and induce H_2_S -generating enzymes. Another possibility is that a part of the H_2_S bioequivalents found in blood and tissues of conventional mice is produced by sulfur-metabolizing bacteria in the gut lumen [122]. In addition, the characteristics and composition of the individual diet may have an impact on the concentration of luminal sulfide in the large intestine.

As mentioned earlier, H_2_S has dual effects in colonocytes that depend on the specific concentration of the gasotransmitter (see also Table 2). Low concentrations of H_2_S are anti-inflammatory, whereas elevated concentrations of H_2_S exert pro-inflammatory effects. A number of studies reported that physiological concentrations of microbial H_2_S production are beneficial for the host [16,122,123]. Critical evaluation of available findings includes both pro- and anti-inflammatory actions of H_2_S in the intestinal tract, as well as smooth muscle relaxation, pro-secretory and pro- and anti-nociceptive effects [124].

The regulation of H_2_S concentration in the intestine depends on the balance between the activity of bacterial inorganic respiration, fermentative bacteria and sulfur amino acid metabolism by colonic cells themselves [16]. The co-evolution of the microbiota and the intestinal epithelial cells enables a sulfur metabolic system which maintains H_2_S at a constant concentration relying on a predominantly anaerobic microbiota and an aerobic colonic cell [16]. Findings by Linden et al. support the proposal that high expression of *SQOR* in the colonic mucosa ensures rapid oxidation of H_2_S, which would substantially limit or preclude gasotransmitter transfer into the bloodstream [125]. The complexity of host–microbiome processing of H_2_S has been reviewed in detail [124] and calls for further investigation.

### 2.4. Catabolism of H_2_S

Catabolism of hydrogen sulfide occurs primarily via the oxidation pathway in the mitochondrion with the concomitant production of ATP [126]. Oxidation of H_2_S takes place in a variety of mammalian tissues including liver, kidney, heart and colonocytes. At the molecular level, two H_2_S molecules diffuse into the mitochondria and bind to the active site of two *sulfide quinone oxidoreductase (SQR*) enzymes [66], each of them reducing two cysteine disulfide, forming a sulfane-sulfur group (*SQR*-SSH). In the next step, the sulfane-sulfur group is transferred to either glutathione, dihydrolipoic acid or to Trx. Electrons derived from H_2_S oxidation are transferred via ubiquinone to the respiratory chain, where they reduce coenzyme Q via the cofactor FAD [126,127] and drive ATP synthesis [128]. Glutathione persulfide (GSSH) is further oxidized by *persulfide dioxygenase* (*SDO*) (encoded by the gene *ETHE1*) to sulfite, which is oxidized to sulfate by downstream *sulfite oxidase* [108,129,130]. Alternatively, sulfite can be metabolized by *thiosulfate sulfurtransferase* to thiosulfate [131]. The enzyme *rhodanese* has been recognized in the mitochondrial oxidation pathway. According to recent studies, *rhodanese* catalyzes the transfer of sulfane sulfur from GSSH to sulfite to form thiosulfate [64,65]. *Rhodanese* is localized in liver, kidney [132] and also in colon, where it seems important in the detoxification of H_2_S produced by the gut microbiota [133,134]. In summary, *SQR* and *sulfur dioxygenase* are reported as possible key regulators of H_2_S catabolism [66]. They might switch off sulfide signaling by consuming H_2_S and its product (Figure 7). 

Under oxidative conditions, a fraction of H_2_S can be scavenged by coordination to the heme-Fe(III) atoms of methemoglobin and metmyoglobin [135,136] to form sulfhemoglobin (SulfHb) [128]. In mammals, metallo- and disulfide-containing macromolecules such as horseradish peroxidase and catalase and oxidized GSH also scavenge H_2_S [137,138]. Another mechanism for H_2_S clearance is its complexation with other molecules (e.g., perthiols and nitrosothiols) that undergo removal by diffusion across respiratory or epithelial surfaces through metal- and enzyme-catalyzed oxidation [111,137].

### 2.5. Reservoir of H_2_S

Because H_2_S and its metabolites (such as inorganic and organic persulfides and polysulfides, thiosulfate or metal-coordinated sulfur species) are highly reactive molecules [59], and sulfur species can spontaneously interconvert via fast dynamic equilibria, a dynamic system for controlled H_2_S storage and release appears beneficial [26]. 

Evidence exists that H_2_S could be stored and released as a response to a physiologic signal [38,139]. Two forms of sulfur stores in cells have been identified [140], namely, acid-labile sulfur, which releases H_2_S under acidic conditions, and bound sulfane sulfur, which releases H_2_S under reducing conditions [139]. Acid-labile sulfur consists of sulfur atoms in the iron–sulfur complexes, which play an important role in a wide range of redox reactions in enzymes of the respiratory chain in the mitochondria [139]. The critical pH below which H_2_S is released from acid-labile sulfur is 5.4 [38]. Thus, this reaction is unavailable as a source of H_2_S inside the mitochondrion, where pH > 7.8. Exogenously applied free H_2_S is immediately absorbed and saved as bound sulfane sulfur, suggesting that probably enzymatically produced H_2_S as described above may also be stored as bound sulfane sulfur [38]. In particular, H_2_S produced by *3*MST was reported to be stored as bound sulfane sulfur, which influences the intracellular bound sulfane sulfur concentration [105]. In turn, cells may release the stored H_2_S after receiving a certain physiological signal, such as elevated pH [105]. Experimentally, the reducing activity of thiols such as GSH and cysteine is greater in alkaline conditions than at a neutral pH. Therefore, most H_2_S release from lysates of cultured neurons and astrocytes occurs at a pH higher than 8.4 [38]. 

### 2.6. Antioxidant Functions of H_2_S

As is the case with other gasotransmitters, H_2_S exerts its functions by binding to biomolecules in a transient way as well as in a stable fashion via modification of their thiol groups. H_2_S can exert antioxidant functions by induction of antioxidant pathways such as Keap1/Nrf2 that control glutathione biosynthesis [141,142,143], by persulfidation of cysteine residues in proteins [144,145,146], by acting on the mitochondrial respiratory chain to control ATP synthesis [147] and by direct reaction with metal centers [148], such as hemoglobin [116] and cytochrome c oxidase [47,149]. Direct reactions of H_2_S with reactive oxygen species also occur, but given the low concentrations of H_2_S in cells and tissues, this is unlikely to be a major contribution to overall antioxidant defense [150]. A recent comprehensive in vitro study identified nutraceutical polyphenols (berry extracts) that oxidize H_2_S to potent antioxidant polysulfides that initiate cytoprotective effects [151]. The mechanism suggest that polyphenols undergo auto-oxidation to their corresponding semi-quinone species, and in turn react with H_2_S to form thiyl radical and ultimately polysulfides and thiosulfate [151]. Increasing knowledge of the biochemistry of H_2_S may be exploited for the design of dietary intervention studies aimed at augmenting antioxidant defense. Detailed reviews on the roles of H_2_S in antioxidant functions with potential clinical relevance are available [150,152,153,154]. A review of chemical tools to study small molecule persulfides has also been published [155]. 

## 3. Methods of Detection of Hydrogen Sulfide

Because sulfur is a physicochemical and geological indicator, and hydrogen sulfide is a natural and anthropogenic pollutant [156], the first methods to detect H_2_S were developed in the field of environmental chemistry, hence not suited for the analysis of the gasotransmitter in human and other animal samples. 

As H_2_S gained appreciation as a signaling molecule with important roles in human health, the development of methods to detect and quantify the gasotransmitter became a necessity and an exciting subfield of research. Methods based on fluorescence, polarography and chromatography were developed over the years, each carrying intrinsic advantages and disadvantages [156]. A summary of most commonly employed methods for the detection of H_2_S in biological compartments is provided in Table 3. The accurate determination of H_2_S concentrations in biological specimens remains a matter of active investigation. The major experimental challenge concerns the great reactivity of H_2_S with numerous low and high molecular weight targets. As can be appreciated from the summary in Table 3 and the following sections, discrepancies still remain about what concentrations are free from unintended experimental artifacts, and which genuinely represent physiological ranges of H_2_S concentration. 

### 3.1. Direct UV-Visible Spectrophotometric Measurement

Two major spectrophotometric methods have been described for the measurement of H_2_S: one direct, by looking at the absorbance of sulfide species in the UV region of the spectrum (214–300 nm) [157], and an indirect method that utilizes the H_2_S-binding hemoglobin from the marine organism *Lucina pectinata*, known as hemoglobin I [158].

The direct spectrophotometric method enables the determination of total dissolved sulfide concentration in natural waters through direct ultraviolet measurement of the HS^−^ ion [157]. Bisulfide concentration can be determined by measuring absorption from 214 to 300 nm and afterwards delimitation of the HS^−^ spectra from the complex spectrum of the fluids [157]. In samples with a lower sulfide concentration, the spectrophotometric method enables simultaneous determination of other UV^−^ absorbing ions such as nitrate, iodide and bromide. In order to have an exact measurement of bisulfide concentrations, samples should have low background absorption. Under these conditions, the detection limit is reported to be less than 1 µM [157]. The advantages of this method are first and foremost simplicity and a fast data acquisition [157]. The optimal pH range for UV detection of total sulfide concentration lies in the range between 8.0 and 9.0 [157]. Alkaline conditions favor HS^−^ formation, so bisulfide is more than 95% of total sulfide. An alkaline solution containing, e.g., 50 µM total sulfide produces a well-defined absorbance peak at around 230 nm [157]. Several modifications of this method have been reported for the analysis of sewage and seawater [159,160,161], but of course, due to the complex composition of human and other animal samples, these approaches are not adequate for the quantification of H_2_S in blood, tissue or other specimens [157,162]. 

The indirect “zinc-trap” method is one of the oldest and most commonly applied techniques for the detection of H_2_S. It is based on the colorimetric detection of the end product methylene blue (MB) [163]. In the first step, sulfide is trapped with a metal (e.g., zinc acetate), followed by an acidification and reaction with N,N-dimethyl-p- phenylenediamine (DMPD). The next reaction, catalyzed by ferric chloride, generates the methylene blue dye that could be measured at 670 nm using a UV-spectrophotometer or a plate reader. Berliner et al. initially described this method with the main aim of measuring sulfide concentration in aquatic samples [163]. A disadvantage of the method and of colorimetric detection in general, is the possible interference of other chromophores present in the sample. Chromatographic separation was described to be a possible solution to this impurity and it could increase the sensitivity of the method [164,165,166]. A further improvement is the replacement of the UV-spectrophotometer detector with a mass spectrometer [167]. The detection limit was described as around 1 μM [26]. Nevertheless, a deviation from linearity was observed at higher sulfide concentrations in the sample [26]. In addition, the methylene blue assay did not result in a well-defined H_2_S peak (660–680 nm) at low physiological concentrations, in contrast to other analytical detection methods (e.g., HPLC detection of sulfide-dibimane) [33,168].

Another indirect measurement of H_2_S via its reaction with hemoglobin I was reported as a novel improvement of the UV-based spectrophotometric method [158]. This method is based on the high affinity and high specificity 1:1 coordination of H_2_S and the ferric derivative of hemoglobin I (HbI) from the bivalve mollusk *Lucina pectinata* [158] (Figure 8a). The UV-visible spectrum of ferric HbI changes markedly upon binding to H_2_S (Fe(III): 407–410 nm; Fe(III)-SH: 419–425 nm, Figure 8b), permitting the direct quantification of the gasotransmitter in solution at concentrations as low as 1 × 10^−6^ M [158,169]. Ferric *L. pectinata* HbI can be purified directly from the clam or expressed in recombinant form in *E. coli* [170,171,172]. In both cases, HbI is utilized as a colorimetric probe to determine the concentration of pure, commercially available H_2_S. The concentration of H_2_S is determined using the molar extinction coefficient for the sulfide-HbI complex Δε_tot_ = 69.0 M^−1^ cm^−1^) [158]. 

### 3.2. Chromatography

A variety of chromatographic detection methods are available for the quantification of H_2_S and, more generally, of sulfur pools [25]. The sections that follow summarize the most widely used chromatography-based methodologies and their applications. 

#### 3.2.1. Gas Chromatography/Ion Chromatography

Ubuka et al. reported a gas chromatography (GC) method that detects hydrogen sulfide and acid-labile sulfur in fresh rat liver and heart [37]. Hydrogen sulfide is trapped in an alkaline solution and determined by using gas chromatography coupled to a flame photometric detector (GC-FPD). In the gas transfer step, H_2_S is collected from animal tissue samples. Acid-labile sulfur is liberated as hydrogen sulfide under acidic conditions [139] and sulfane sulfur is collected as thiocyanate by treating tissue samples with cyanide [173]. In the gas equilibration step, phosphoric acid is added to the specimen. Liberation of H_2_S from the aqueous phase into the gas phase in the head space of a closed vial is induced by stirring of the sample. The head-space gas is then transferred to a gas chromatographer for measurement. According to the study, free hydrogen sulfide and sulfur dioxide could not be detected in the liver and heart with this gas chromatographic analysis. Therefore, they assumed that H_2_S could be detected by ion chromatography after oxidation of hydrogen sulfide by hydrogen peroxide. The detection limit of this second step of sulfate and thiosulfate was reported as 20 pM [37].

#### 3.2.2. Flow Gas Dialysis/Ion Chromatography

The studies of Goodwin et al. described a very similar analytical technique to the above described chromatography for the determination of sulfide in human and rat brain tissue with a detection range of 100 μM [41]. They used continuous flow gas dialysis as pre-treatment of the sample, followed by quantitation by ion chromatography with electrochemical detection. The advantage of this method is a pre-concentration of the analyte before the quantitation step. The continuous flow gas dialysis system separated sulfide from the homogenate matrix. The sample homogenate stream reacted with HCI in order to release the H_2_S into a NaOH receiving solution. The receiving solution is collected in a vial and injected into the ion chromatograph. The electrochemical detector had a silver working electrode, a silver/silver chloride reference electrode and a stainless-steel counter electrode. This method is reported to be very selective and has until now no known interferences. Therefore, it could be a promising technique for the post-mortem analysis of H_2_S concentrations in tissues from individuals who suffered from inhalation poisoning. Nevertheless, sample preparation is labor-intensive and quantification of H_2_S by this method requires specialized instrumentation [41].

#### 3.2.3. Gas Chromatography/Chemiluminescence Sulfur Detector

The combination of the described GC method with a chemiluminescence sulfur detector was reported to detect free H_2_S concentration in liver and brain tissue of mice and human blood [32]. The detection limit is in the nM range [32]. A small gas sample is injected into a Teflon column packed with Chromosil 330, and nitrogen gas is used as a carrier with a flow rate of 25 mL/min. The concentration of H_2_S could be determined via comparison of the peak areas of the unknown to those of known commercial standards. According to recent studies, very low concentrations of free H_2_S could be measured in mouse brain tissue (14 nM), liver tissue (17 nM) and in human blood (100 pM) [32]. Levitt et al. used a different chemiluminescence sulfur detector and they were able to determine a concentration of H_2_S in mouse blood of 15 nM [34]. These findings highlight the marked variation introduced by different instrumentation and possibly by modification of the sample preparation step.

#### 3.2.4. Gas Chromatography with Silver Particles

Ishigami et al. developed a GC-based method using silver particles to trap free H_2_S from rat brain tissue homogenate [38]. The detection limit of this method was reported to be 9 µM [48]. Small particles of silver absorb H_2_S as Ag_2_S from the biological sample. Proteins that adhere to silver particles are removed with Triton X100. Next, thiourea and H_2_SO_4_ are applied to release H_2_S from silver sulfide produced on the surface of the particles. This procedure excludes the contribution of acid-labile sulfur. Nevertheless, no H_2_S was detected because the detectable concentration in the brain is ~9.2 µM [38]. This result showed that under basal physiological conditions, free H_2_S in the brain is maintained at a very low concentration. The concentration of H_2_S released from homogenates of whole brain, liver and heart was measured at various pH values. The authors reported that the critical pH to release H_2_S from acid-labile sulfur was ~5.4 [38].

In sum, a major disadvantage of the above described variations of the chromatography method are the multi-step sample preparation and the relatively long incubation times that are needed for gas evolution into the headspace of the collecting chromatography sample vial. These aspects pose a significant limitation, especially for real-time measurements of H_2_S production [128]. These methods detect, first and foremost, H_2_S gas alone instead of other bioavailable forms of sulfide (e.g., HS^−^, acid labile sulfide and bound sulfane sulfur). In that respect, these approaches are sensitive and specific for H_2_S [26].

#### 3.2.5. High Pressure Liquid Chromatography (HPLC)

The use of HPLC by several laboratories has permitted the development of methods that measure bioavailable sulfide in various forms. Savage and Gould reported on the derivatization of sulfide and separation by reversed-phase HPLC (RP-HPLC) coupled to spectrophotometric detection of methylene blue [174]. Zinc acetate was utilized to trap and determine H_2_S concentration in acidic medium. Ogasawara et al. improved this HPLC method through pre-column fluorescence derivatization to detect trace amounts of sulfide [175]. After detection, those small sulfide quantities were converted into thionine, a fluorescent derivative. In a further step, sulfide reacted with N,N,-dimethyl-p-phenylenediamine to methylene blue. This method has a detection limit in the micromolar range [175]. The HPLC method eliminates interferences that are problematic in spectrophotometric methods such as turbidity of the sample, the presence of impurities and the formation of colored byproducts [24].

Kožich et al. described an indirect measurement of H_2_S biosynthesis by liquid chromatography–mass spectrometry (LC-MS/MS) [73]. The method examined end-point concentrations of the thioethers cystathionine (Cysta), homolanthionine (HLH) and lanthionine (LH) as fingerprints of H_2_S production in reactions driven by CBS. These thioethers are stable surrogate markers for H_2_S biogenesis (Figure 4). The study determined thioether concentration in human plasma and in fibroblasts from patients with CBS deficiency, remethylation defects or nutritional vitamin B12 deficiency and compared them to the corresponding assessment in specimens from healthy controls. The detection of analytes was carried out using a commercially available kit for amino acid analysis, with mass spectrometry conditions set to positive electrospray ionization and selected multiple reaction monitoring. The detection limits for all three thioethers were approximately 1 nmol/L. 

### 3.3. Monobromobimane Assay

According to Wintner et al., the concentrations of reactive sulfide in saline and freshly drawn whole blood could be quantified through reactions with the thiol-specific derivatization agent monobromobimane (MBB), followed by reversed-phase fluorescence HPLC and/or mass spectrometry [35]. Monobromobimane has a high affinity for low molecular weight thiols [35]. This chemical probe undergoes nucleophilic substitution with sulfide, wherein two equivalents of MBB react with sulfide to form the highly fluorescent sulfide dibimane (Figure 9). As bimane itself is relatively hydrophobic, sulfide dibimane is even more hydrophobic than most monobimane derivatives of physiological thiols. Therefore, the sulfide dibimane species can be easily separated from other thiol bimane derivatives by reversed-phase chromatography (RP-HPLC). 

In order to quantify the formation of sulfide dibimane in a sample, the next step is RP-HPLC with monitoring of UV absorption (254 nm) and fluorescence (lex = 390 nm, lem = 475 nm). In the studies by Wintner et al., formation of sulfide dibimane was followed by monitoring fluorescence intensity over 5 min [35]. 

Wintner et al. optimized assay conditions for the efficient and reproducible reaction of MBB with sulfide to form sulfide dibimane [35]. The assay is performed at room temperature, at pH 8.0, and has a detection limit of 0.7 µM free H_2_S in rat blood [35]. Nevertheless, it is important to mention that the MBB assay measures all bioavailable sulfide, in the gas as well as in the anion form (H_2_S/HS^−^). Since both forms are biochemically active, this assay quantifies H_2_S bioequivalents relevant to physiology.

An improved protocol based on MBB with RP-HPLC and fluorescence detection was developed to determine H_2_S speciation [33]. The method comprises selective liberation, trapping and derivatization of H_2_S. In the first step, acid-labile H_2_S is liberated by incubating the sample under acidic conditions. Volatilized H_2_S is then trapped and added to an excess of MBB. In a separate step, bound sulfane sulfur is incubated in the presence of the reducing agent TCEP. H_2_S is again trapped and derivatized with MBB. Separation and quantification of the species by RP-HPLC and fluorescence detection were analogous to the protocol described above. Finally, the acid-labile pool was calculated by subtracting the concentration of free H_2_S from the concentration of sulfide obtained by acid liberation. The bound sulfane sulfur pool was determined by subtracting the concentration of H_2_S from the acid-liberation step alone from that of TCEP plus acidic conditions. 

While the MBB method seems like an accurate, quantitative and scalable measurement of discrete pools of hydrogen sulfide from primary volatile sulfide pools, it is not free of limitations. Apart from the multiple experimental steps required to measure the concentration of H_2_S both in solution and in its gaseous phase, carefully controlled studies of reactivity revealed that MBB itself modifies sulfur speciation in biological specimens such as plasma and serum [26,59]. The concentration of sulfide determined by the MBB method in serum samples was found to depend on incubation time and on the concentration of MBB itself [59]. These findings are in line with an independent report that compared sulfur quantitation side-by-side using MBB, iodoacetamide (IAM) and N-ethylmaleimide (NEM) [176]. The readout of sulfide concentration in whole blood was higher with probes NEM and MBB compared to the same measurement utilizing IAM [176]. These differences can be explained by a greater capacity of NEM and MBB in cleaving polysulfur chains compared to IAM. These limitations have lead experts in the field to recommend “extraordinary care” with the reporting of “free sulfide levels” and with the assignment of roles to specific sulfur species in biological systems [59]. 

### 3.4. Other Fluorescent Probes for H_2_S Detection

Aside from the MBB-based methods, variants that employ other fluorescent probes have also been described [177]. One example is the use of benzodithiolone to determine the concentration of H_2_S in plasma [177]. In order to detect H_2_S selectively, the main aim was to distinguish H_2_S from other biological nucleophiles, especially thiols such as glutathione and cysteine. In contrast to the other thiols, H_2_S can undergo nucleophilic reaction twice. Cys and other thiols are monosubstituted and can therefore undergo the nucleophilic reaction only once. Benzodithiolone has a unique bis-electrophilic center, which could be a useful reagent for H_2_S detection. Using this strategy, the concentration of H_2_S could be measured in two ways: firstly, by following the fluorescent signal, and secondly, by the analysis of the benzodithiolone product (by mass spectrometry, for example).

Studies by Peng et al. described dansyl azide (DNS-Az), a novel reduction-sensitive, stable and non-fluorescent chemoprobe [178]. The reagent became fluorescent through reacting with sulfide. The reaction took place very rapidly, enabling the detection of transient changes in H_2_S concentrations without any sample pre-treatment. Furthermore, the probe was simple in structure, very easy to synthesize and amenable to long-term storage.

Work by Thorson et al. described the utility of 7-azido-4-methylcoumarin (AzMC) for monitoring H_2_S production by enzymatic sources and screening for potential inhibitors [179]. This fluorogenic probe presented a dynamic range of detection of 0.1–100 µM and did not react with transsulfuration enzyme substrates Cys or Hcy at concentrations of 10 mM nor with 1 mM PLP or S-adenosylmethionine. However, the authors noted interference with the commonly used reducing agents DTT and TCEP. AzMC did not react with glutathione or 2-mercaptoethanol [179]. 

Another new fluorescent probe for H_2_S detection is 4-methyl-2-oxo-2H-chromen-7-yl 5-azidopentanoate (a self-immolative coumarin-based fluorescence probe) [40]. The design of this probe was based on the combination of two characteristic reactions performed by H_2_S: firstly, the reduction of an azido group to an amine by H_2_S, and secondly, a spontaneous intramolecular lactamization, concomitant to the formation of highly fluorescent 7-hydroxy-4-methylcoumarin (also known as 4-methylumbelliferone and hymecromone). The advantages of this probe were described as follows: the synthesis and purification steps were straightforward, the substance reacted rapidly (within seconds, end-point imaging of H_2_S -formation could be possible), the substance was chemically stable which could enable a long-term storage, it showed a linear concentration–signal relationship at relevant H_2_S concentrations and it was stable in aqueous solutions, especially at physiological pH. The optimal pH was described as around ≈ 7.0, which is comparable with the pH of many mammalian body fluids. The benefit of this substance in comparison to other fluorogenic assays was that self-immolative probes generate a more stable signal with a higher signal-to-noise ratio [40].

Dulac et al. described a selective and reversible fluorescent system to detect hydrogen sulfide [156]. The competitive detection limit was claimed to be ∼200 nM at pH 7.4 [156]. Firstly, they looked for an avid biological system for hydrogen sulfide as a sensor. This unique characteristic could be found in hemoglobin HbI from the clam *Lucina pectinata* [169]. As described above, the high affinity to HbI for H_2_S could be explained by the special structure of its distal active site which contains the amino acids glutamine and two phenylalanines [135]. Therefore, HbI appeared to be perfectly suited for the biomolecular gating approach. In the first step, Dulac et al. developed a turn-on fluorescent sensor in order to detect H_2_S, with N-[2(2-hydroxyethoxy)ethyl]-6,8-difluoro-7-hydroxycoumarin-3-carboxamide, a derivative of 6,8-Difluoro-7-hydroxycoumarin-3-carboxylic acid, the latter known as “Pacific Blue” [180]. Afterwards, increasing concentrations of rHbI were added to the chemical fluorophore, which resulted in a decrease in fluorescence intensity. Addition of the H_2_S source sodium hydrosulfide (NaSH) restored the fluorescence of the fluorophore, which was indicative of a direct reaction between the derivative of the coumarin dye Pacific Blue and H_2_S. Indeed, after the reaction took place, Dulac et al. observed a red-shift of the Soret band from 407 to 425 nm, which is characteristic of the Hb-SH complex. This result showed a measurable modulation of the photophysical characteristics of a fluorophore with an appropriate excitation around 407 nm. The presence of the fluorophore led to faster coordination of H_2_S to rHbI and a faster dissociation as well with respect to so far published data. In particular, the dissociation measurements revealed a discrepancy with previously reported rates (half-life around 92 s [156], previously reported around 600 s [169]). Unfortunately, the measurement of H_2_S in human plasma was not possible, which indicated that H_2_S concentration in plasma samples could be lower than the quantification limit.

In summary, these fluorescent H_2_S detection methods appear simple and cost-effective. Possible limitations may exist in terms of sensitivity (micromolar) and the time needed to sufficiently measure changes in H_2_S concentration (minutes to an hour). Finally, this technical approach does not detect sulfide concentrations from different biochemical pools within the sample or the fraction of H_2_S bound to proteins [33,168].

### 3.5. Sulfide Specific Ion-Selective Electrodes (ISE)

In these methods, the measurement of H_2_S concentration is performed against a glass pH electrode, upon conversion of all sulfide to S^2−^. The resulting electromotive force depends on sulfide concentration and on the pH of the solution as well [181]. The latter is kept constant and alkaline, to ensure full conversion of all sulfide species to S^2−^. A calibration curve made with commercially available sources of H_2_S is utilized for quantification.

Whitfield et al. demonstrated an erroneous effect of ISEs using trout plasma and bovine serum albumin (BSA) samples [182]. The authors observed that sulfide concentration in their samples increased rapidly over time because of alkaline conditions. Searcy and Lee reported in an earlier study the detection limit to be around 0.5 μM and claimed that the electrode could work accurately both in oxygenated and deoxygenated buffers [110]. Searcy and Peterson demonstrated the sensitivity of this method, maintaining a linear response in a very low (1 fM–1 μM) concentration range [183]. Further described advantages of the method were as follows: it measured free sulfide concentration without a previous sulfide derivatization and it enabled a dynamic measurement due to fast electrode response. Nevertheless, the risk for erroneous measurements is still substantial, most likely because of harsh sample pre-treatment and because of an interference with the Ag^+^/Ag_2_S system on the electrode surface [25,26].

### 3.6. Polarographic Electrodes

Polarographic electrodes are reliable for end-point H_2_S gas detection in biological samples [36]. The detection limit is reported to be in the nM range under anoxic conditions. Koenitzer et al. developed a novel polarographic hydrogen sulfide sensor (PHSS) [36], which is able to record H_2_S concentration in a respirometer chamber or in vessels [184]. The design contained an anode, cathode and a H_2_S-permeable polymer membrane. The PHSS has high affinity for H_2_S, responds quickly to varying H_2_S concentration and has a detection limit of approximately 10 nM [36]. The PHSS provided satisfactory response to the kinetic profile of H_2_S metabolism, which enabled measurements in cell lysates, cell suspensions, intact tissues and whole organisms. Due to high selectivity, PHSS could be combined with other end-point polarographic sensors, e.g., for O_2_ (POS) and NO (PNOS), in order to detect a possible interaction of these factors with H_2_S in biological systems. With such a multisensor respirometry, Doeller et al. demonstrated end-point formation and consumption of H_2_S by mammalian tissues and cultured cells [36].

A recent report by Faccenda et al. showed that polydimethylsiloxane (PDMS) membranes were permeable to H_2_S and could therefore be useful for continuous H_2_S determination [185]. In addition, 96-well inserts constructed of PDMS functioned as a ∼100 μm thick H_2_S-permeable membrane, which was not permeable to other thiols. Therefore, nonspecific thiols could simply be eliminated. The group reported a detection limit of around ∼0.51 μM for free H_2_S (as solution sulfide) [185]. Although this H_2_S polarographic sensor method is sensitive and accurate, it does not detect other biochemical forms of sulfide (e.g., acid labile and bound sulfide). Nevertheless, the PDMS method seems to be the best option for end-point detection of H_2_S gas formation.
antioxidants-10-01065-t003_Table 3Table 3Methods of detection of hydrogen sulfide.MethodSample Pre-AnalyticsLimit of DetectionType of SampleLiterature**1. Direct spectrophotometric measurement**



**1.1. Direct ultraviolet** detection of the HS^−^ ion, absorption measurement from 214 to 300 nm (peak around 230 nm)Sulfide dissolved in natural water, pH ~8.0<1 µMDissolved sulfide in natural waters[157]**1.2. Zinc trap/methylene blue:**Trapping sulfide with a metal (e.g., zinc acetate) with subsequent acidification and reaction with DMPD, formation of methylene blue by a ferric chloride catalyzed reaction, spectrophotometric measurement of methylene blue, improvement by replacing the spectrophotometer detector with a mass spectrometerNo pre-treatment of sulfide is necessary1 μM50 ng L^−^Aquatic samplePSI particles[163,164,165,166,167]**1.3.** Hemoglobin I. Based on the 1:1 reaction between H_2_S and the ferric species of **hemoglobin I** (HbI),simple and fast, measuring of absorbance at 408 nm and 429 nm, determination of the fractional saturation of HbI-H_2_S complexPurification of ferric *L. pectinata* HbI from the clam, determination of HbI concentration, H_2_S of analytical grade was used without further purification, calculation of solubility of H_2_SUp to nMFerric *L. pectinata* HbI, dissolved H_2_S of analytical grade[158,169]**2. Chromatography**



**2.1. Gas chromatography/ion chromatography:** Determination of H_2_S using an A7 column equipped with a flame photometric detector (GC-FPD)/quantitation with electrochemical detectionTissues are treated with acid to liberate H_2_S from ALS, sulfane sulfur is liberated as thiocyanate by treating tissues with cyanide, homogenization with sodium chloride/oxidation of hydrogen sulfide by hydrogen peroxide>100 nm/gRat liver and heart tissues[37]**2.2. Flow gas dialysis/ion chromatography:** Separation of sulfide from the homogenate matrix enables a pre-concentration/electrochemical detection and quantitation of H_2_S through a silver working electrode, a silver/silver chloride reference electrode and a stainless-steel counter electrodeFatal intoxication of the brain by intraperitoneal injection of NaHS, homogenization with NaOH, reaction of the homogenate with HCl to release H_2_S100 μMHuman and rat brain tissue[41]**2.3. Gas chromatography/chemiluminescence:** Injection of a gas sample onto a Teflon column, determination of H_2_S concentration via comparison of peak area of the unknown to that of known standardsIncubation of homogenized tissue in media, homogenization in ice-cold potassium phosphate buffer, addition of aliquots of the tissue homogenate to pre-warmed polypropylene syringes17 nM14 nM100 pMLiver tissueBrain tissue of miceHuman blood[32]**2.4. Gas chromatography/silver particles:** Adsorption of H_2_S as Ag_2_S through silver particles from the sample, application of thiourea and H_2_SO_4_ to protein-free silver particles to release H_2_S from silver sulfide produced on the surface of the particles, measurement of H_2_S concentration at various pH values with GCAlkalinization of the cytoplasm for H_2_S release from bound sulfur, acid-labile sulfur: homogenization of rat tissues with NaOH, neutralization of cleared supernatant with phosphoric acid, bound sulfur: homogenization of rat or mouse tissue with a polytron homogenizer in ice-cold buffer, addition of DTT to supernatant15 nM9 μM6 pMMouse bloodRat brainRat homogenate tissue[34,38]**2.5. Liquid chromatography–mass spectrometry:** Measurement of CBS activity, using the stable isotope substrate2,3,3-2H serine, monitoring of 3,3-2H-cystathionine, cystathionine and 3,3,4,4-2H-labeled cystathionine formationAddition of pyridoxine to all plasma samples at the time of sampling, addition of plasma or serum to a solution containing Tris- pyridoxal 5′-phosphate and 2,3,3-2H-labeled serine, activation of CBS by SAMUp to nMHuman plasma or serum[186]**2.6. Modification of liquid chromatography–mass spectrometry:** Optimization of liquid chromatography with a commercially available Symmetry C18 column and MS/MS detectionNo difference0.1 μMHuman plasma or serum[187]**2.7. HPLC:(2.7.1)** Derivatization of sulfide via reversed-phase HPLC (RP-HPLC) separation, determination of sulfide via spectrophotometric detection of methylene blue, trapping of H_2_S with zinc acetate, measuring of H_2_S concentration under acidic conditionsAddition of zinc acetate to each rumen fluid sample, homogenization of cerebrocortical gray matter in aqueous zinc acetate, treatment of human whole blood with heparin, preparation of spiked samples by adding sulfide standard solutions, conversion into the fluorescent derivative thionine, under acidic conditions0.123 to 0.189 pmol dm^−3^Bovine brain tissue[174]**(2.7.2)** Separation of sulfide on a reversed-phase column and fluorimetrical detection, after pre-column fluorescence derivatization, conversion of low sulfide concentration into thionine, formation of methylene blue

Human red blood cells[175]**3. Monobromobimane assay (gold standard)**



**3.1.** Reaction with the thiol-specific derivatization agent monobromobimane to sulfide-dibimane, detection of sulfide-dibimane concentration by reversed-phase fluorescence HPLC and/or mass spectrometryReaction of sulfide with 2.0 equivalents of monobromobimane, formation of sulfide-dibimane0.7 μMRat bloodHuman plasma[35]**3.2.** Monobromobimane method coupled with RP-HPLC and fluorescence detection, measurement of sulfide by methylene blueLiberation of acid-labile H_2_S by incubation of the sample in an acidic solution (pH 2.6) with diethylenetriaminepentaacetic acid (DTPA), incubation of the sample with TCEP0.7 μMMouse plasma[33]**4. Fluorescent probes**



**4.1.** Conductometric **gas nanosensors** for H_2_S, which are synthesized by electrodepositing gold nanoparticles on single-walled carbon nanotube (SWNT) networks, detection of H_2_S is based on conductivity changes of thin films upon exposure to H_2_S gasHydrogen sulfide (purity: 99.998%) diluted in dry air at a gas flow of 200 std. cm~3 min^−1^3 ppbH_2_S gas[188]**4.2.** Measurement of H_2_S plasma concentration by assessing the fluorescence signal from **benzodithiolone** product formation. Benzodithiolone has a bis-electrophilic center with specificity for H_2_S detectionPreparation of the probe with thiosalicylic acid, preparation of bovineplasma containing NaHS at different concentrations (0, 50, 100 and 500 mm)
Reactive disulfide-containing probeBovineplasma[177]**4.3. Dansyl azide (DNS-Az)** is a reduction-sensitive, stable, non-fluorescent chemoprobe, it becomes fluorescent upon reacting with sulfideNo sample pre-treatment1 μM5 μMAqueous solutionBovine serumMouse blood[178]**4.4. 7-azido-4-methylcoumarin (AzMC)** is a fluorogenicprobe selective for H2S; it provides a facile, direct, continuous and sensitive assay for activity monitoring of PLP-dependentenzymesSynthesis and purification of hCBS0.1 µMTruncated version of human CBS (without the regulatory domain)[179]**4.5. 4-methyl-2-oxo-2H-chromen-7-yl 5-azidopentanoate,** a fluorescent probe for H_2_S detection, two characteristic reactions of H_2_S: reduction of an azido group to an amine, intramolecular lactamization with simultaneous release of highly fluorescent 7-hydroxy-4-methylcoumarinSaliva samples were tested instantly, measurements do not require any chemical pre-treatment, aqueous medium at pH = 7.41.641–7.124 µMSodium phosphate buffer at physiological pHSaliva[40]**4.6. N-[2(2-hydroxyethoxy)ethyl]-6,8-difluoro- 7-hydroxycoumarin-3-carboxamide** is a turn-on fluorescent sensor to detect H_2_S, the **sensor rHbI** functions as a filter at the excitation wavelength of fluorophore in absence of H_2_S, but not in its presence, permits end-point measurement of H_2_S concentration, reversible and selectiveExpression of recombinant rHbI in *E. coli* BLi5 strain, purification of recombinant rHbI, UV−vis spectrum to confirm the formation of rHbI-H_2_S, adding of potassium cyanide to the reaction mixture, monitoring of displacement reaction by UV−vis spectroscopy∼200 nM at pH 7.4Recombinant hemoglobin I from *Lucina pectinata*Human plasma[156,180]**5. Sulfide specific ion-selective electrodes**



Measurement against a glass pH electrode, ISEs measure the HS^−^ form of sulfide in an alkaline environment to favor HS^−^ formationSamples placed in alkaline buffer, saturated solutions of sparingly soluble metal salts1 fM–1 μMBovine serum albumin (BSA)[25,26,183,184]**6. Polarographic electrodes**



**6.1. PHSS** records the concentration in a respirometer chamber/vessel, selective for H_2_S, responds rapidly to varying H_2_S concentrations, contains a H_2_S-permeable polymer membraneInjection of supernatant into the respirometer chamber containing PBS, addition of the substrate L-cysteine and the cofactor PLP to support the activity of enzymes CBS and CSE10 nMRat aortaSmooth muscle cellsIntact rat thoracicaorta[36]**6.2. Polydimethylsiloxane (PDMS) membranes** are permeable to H_2_S, a continuous measurement of H_2_S is possible, not permeable for other thiolsPurification of CSE∼0.51 μMFree H_2_S[185]

## 4. Determination of Hydrogen Sulfide Production in Genetic Diseases of Sulfur Metabolism

Inborn errors of metabolism represent a heterogeneous group of at least 700 rare diseases, which are caused by inherited deficiencies of enzymes, transporters and other gene products [189,190]. Genetic defects in sulfur amino acid metabolism lead to an accumulation of the metabolite homocysteine, due to a blockade in the transsulfuration or remethylation pathways. Many in-depth studies described naturally occurring mutations, protein structures and consequences for metabolism and phenotype. Despite its informative value in terms of residual enzyme activity and the clinical presentation of patients with inborn errors that impair H_2_S-pathways, very few published works determined the concentration of product H_2_S or examined its potential contribution to pathogenesis. A summary of H_2_S concentrations determined in different genetic diseases of metabolism is presented in the sections below, covering both human and animal studies. Main findings from studies where H_2_S was measured are also provided in Table 4. All enzymes described in section B, which take part in the biogenesis and the catabolism of sulfur amino acids, can carry an inborn mutation and therefore lead to homocystinuria. For example, 177 missense mutations have been found in the human CBS gene [191], many of which present decreased activity [192], which could also imply impaired H_2_S homeostasis. 

### 4.1. H_2_S Metabolism in Human Studies

Studies by Kožich et al. determined sulfur species in patients with two different types of homocystinuria [42]. The first group consisted of patients with the severe CBS deficiency (CBSD), who do not respond to intervention with pyridoxine, while the second group included patients with remethylation defects (RMD), who often present with a more moderate form of homocystinuria. In the samples of patients with CBSD, homocysteine and methionine accumulated while metabolites below the CBS block (i.e., cystathionine and cysteine) were decreased. In contrast, patients with RMD exhibited decreased production of methionine and S-adenosylmethionine, an accumulation of Hcy and an increased flux of sulfur compounds through the transsulfuration pathway as indicated by the accumulation of cystathionine. Plasma sulfide concentration was determined using monobromobimane (MBB) derivatization followed by HPLC separation and fluorescent quantitation of the sulfide dibimane (SDB) product as described in previous publications [33,35]. This study employed two slightly different experimental settings in order to measure different fractions of the bound sulfide pool (based on the use of different MBB concentrations, which affects the kinetics of sulfide release) beside free sulfide. Both methods indicated that sulfide concentration in plasma of CBSD patients (0.65 µM L^−1^/0.15 µM L^−1^) did not differ from those of the control PKU (phenylketonuria) group (0.5 µM L^−1^/0.19 µM L^−1^). Surprisingly, the measured median sulfide concentrations in plasma of patients with RMD were reported to be significantly decreased to 53% and 64% of the median in healthy controls (RMD: 0.32 µM L^−1^/0.12 µM L^−1^, controls: 0.6 µM L^−1^/0.2 µM L^−1^) [42]. One possible interpretation of these results is that patients with CBSD may have produced slightly less H_2_S from Cys, which was compensated by production of H_2_S from Hcy by CSE and other enzymes, and that this slightly higher H_2_S is further metabolized to thiosulfate.

Studies on H_2_S production with variant p.P49L of human CBS demonstrated a 3-fold reduction (0.9 mol H_2_S mol CBS^−1^ min^−1^ versus 0.3 mol H_2_S mol CBS^−1^ min^−1^) of H_2_S production compared to wild type CBS [193]. The concentration of H_2_S was measured either by amperometry using a H_2_S-selective electrode or by the lead acetate method [97]. Additionally, they wanted to evaluate by UV-visible absorption spectroscopy the impact of this mutation on the protein redox spectra. First, they measured the H_2_S production of the canonical CBS reaction with Cys and Hcy as substrate in the absence of PLP (PLP-“untreated”). Under these conditions, the basal activity of the p.P49L mutant was more than 3-fold lower than that of the WT enzyme. In contrast, addition of SAM resulted in similar readouts of H_2_S synthesis in WT and mutant p.P49L. Secondly, Vicente et al. purified the variant in the presence of PLP (PLP-“treated”) and reported CBS activity similar to the WT enzyme, despite presenting impaired activity stimulation by SAM [193]. Furthermore, they analyzed the structure of the p.P49L variant in detail and discovered a higher flexibility in the region harboring both heme ligands, which could provide a possible increased affinity of CBS p.P49L for CO. 

Because the major cellular H_2_S-producing enzymes utilize PLP as a cofactor, the nutritional status of vitamin B6 may influence H_2_S production in humans. DeRatt et al. investigated this relationship using the gas chromatography method with a sulfur chemiluminescence detector and performed a quantitative analysis of surrogate markers of H_2_S production lanthionine and homolanthionine [194]. Healthy adults undertook a 28 day dietary vitamin B6 restriction and plasma concentrations of lanthionine and homolanthionine were measured at different time points. DeRatt et al. reported no significant effect of vitamin B6 restriction on mean lanthionine and homolanthionine concentrations [194]. The precursor–product relation between cysteine and lanthionine was preserved before and after vitamin B6 restriction, which suggested that lanthionine production by CBS was not affected by short-term vitamin B6 insufficiency. In contrast, the precursor–product relation of Hcy and homolanthionine was diminished after vitamin B6 restriction, likely because of the reduction in CSE activity caused by vitamin B6 insufficiency. These findings suggest that homolanthionine production could be associated with vitamin B6 status in humans, although a moderate short-term deficiency did not change plasma homolanthionine concentrations significantly [194]. 

While transsulfuration reactions are intracellular, CBS activity has been detected in human plasma [186], and CBS and D-amino acid oxidase protein were detected in human cerebrospinal fluid by Western blot [195]. The finding of plasmatic CBS enzymatic activity has been proposed as a tool for the diagnosis of patients with classic homocystinuria [186]. The assay employs only 20 µL of plasma and utilizes isotopically labeled serine (2,3,3-^2^H serine) as substrate to produce the corresponding isotopically labeled cystathionine (3,3-^2^H-cystathionine) as product. Formation of 3,3-^2^H-cystathionine was monitored by LC-MS/MS, and was proven to discriminate clearly between healthy controls and patients with CBS deficiency [186]. 

The *SQQR* enzyme is part of the H_2_S catabolic pathway and mutations in this enzyme might be a new genetic cause of Leigh’s disease [66,131]. Friederich et al. determined H_2_S concentrations in patients with mutations in SQQR [131]. *SQQR* is located in the inner mitochondrial membrane and it oxidizes H_2_S in the first catabolic step (see Figure 7). The study examined the homozygous c.637G > A and c.446delT mutation in different groups of patients. For H_2_S determination, Friedrich et al. used the UV-spectrophotometry method [196]. Functional assays showed impaired *SQOR* enzyme activity due to protein instability as well as isolated decreased complex IV activity, but normal complex IV protein concentration and complex formation. These enzyme variants limited H_2_S metabolization and resulted in episodic hydrogen sulfide accumulation (concentrations not shown) and complex IV inhibition (activity in liver: 0.4 nmol min^−1^ mg protein^−1^ vs. control: 1.8 nmol min^−1^ mg protein^−1^). Elevated concentration of H_2_S inhibits the respiratory chain by coordination of H_2_S to the iron in the heme group of complex IV. 

Kabil and Banerjee reported on two different mutations in *persulfide dioxygenase (ETHE1*), which causes ethylmalonic encephalopathy in humans [197]. This enzyme converts persulfides into sulfite in the mitochondrial sulfide oxidation pathway. The study examined the kinetic activity of human recombinant *ETHE1* variants T152I and D196N. *ETHE1* was described as an oxygen-dependent enzyme, which suggests possible H_2_S accumulation under hypoxic conditions. The study determined thiosulfate and H_2_S concentration via MBB, followed by HPLC and mass spectrometry. Both mutations resulted in a several-fold lower enzyme activity and an accumulation of hydrogen sulfide. Activity findings and concentrations of H_2_S are found in this study. 

Inborn errors affecting *ETHE1* were also examined by Tiranti et al. [197]. This study investigated H_2_S concentration in plasma and urine of patients with diagnosed ethylmalonic encephalopathy, as well as the phenotype of an animal model of *ETHE1* deficiency. The concentration of H_2_S in the samples was determined with the zinc acetate trap method, followed by gas chromatography with a sulfur chemiluminescence detector. The experimental results showed several-fold higher urinary thiosulfate concentration (18 nmol per mg creatinine, 15 nmol per mg creatinine) in both *ETHE1*^−/−^ mice and in individuals carrying mutations in the ETHE1 gene compared to their corresponding healthy control groups (2 nmol per mg creatinine/1 nmol per mg creatinine). These results suggested an increased endogenous production of thiosulfate, which reflects the presence of hydrogen sulfide. In addition, the group identified low cyclooxygenase (COX) activity in muscle and brain, which was attributed to the toxic accumulation of H_2_S. In contrast, they reported normal COX activity and abundance in *ETHE1*^−/−^ liver, despite high H_2_S concentrations. This observation may reflect the presence of organ-specific alternative metabolic pathways for H_2_S detoxification. Ditrói et al. later recapitulated the findings of Tiranti et al., by means of an optimized monobromobimane method [27,197]. 

### 4.2. H_2_S Metabolism in Animal Models of Human Disease

#### 4.2.1. Cystathione β-Synthase (CBS)

Mice carrying homozygous mutations in the CBS gene do not thrive beyond the first 3 weeks of life [198]. Results from a recent study indicate that lethality in CBS knockout mice is brought about by abnormal buildup of methionine in the liver. Accumulation of Met in the liver was accompanied by downstream disturbances in central pathways of energy metabolism [199]. Strikingly, neither a methionine restricted diet nor cystathionine injections could rescue lethality in CBS-null mice [199]. A CBS heterozygous knockout (CBS^+/−^) mouse model was developed by Jensen et al. and the formation of H_2_S in CBS^+/−^ and wild type animals was measured [200]. They chose the methylene blue method in combination with HPLC and described significantly reduced H_2_S concentration by 30–46% compared to wild type in male and female animals (exact concentrations not shown). In a second experiment, the group overexpressed CBS in mouse liver by adenoviral delivery of the human CBS (hCBS) transgene. As a result, the treatment with hCBS adenovirus increased CBS activity by 1.4-fold and the H_2_S concentration in plasma increased by 1.8-fold (50 µM versus 90 µM) [200]. Moreover, Hcy concentration in plasma was significantly reduced. In a third experimental setting, Jensen et al. supplemented mice with ethionine, a methionine analog that activates CBS in the same way as SAM [200]. The results were similar to those described before. In CBS^+/−^ and wild type mice supplemented with ethionine, CBS activity as well as the circulating H_2_S concentration were increased, while Hcy concentration was reduced with respect to mice without supplementation. 

An independent study investigated the phenotype of a heterozygous CBS^+/−^ mouse model [201]. This work assessed CBS enzyme activity and the effect of the long-acting H_2_S donor GYY4137 in wild type under homocystinuric conditions. The synthetic H_2_S donor GYY4137 (crystalline solid in nature) was dissolved in phosphate buffered saline (PBS) and injected intraperitoneally to mice, while the control mice were given normal saline for 6 weeks. The expression of CBS, CSE and MTHFR was decreased in the retina of mice with genotype CBS^+/−^, a result that was reversed upon addition of the H_2_S donor GYY4137 [201]. Treatment of the animals with GYY4137 reduced the concentrations of markers of oxidative stress, which suggests that H_2_S could play an important role in maintaining an antioxidant environment. Additional findings of their experimental work were reduced glutamate concentrations, baseline levels of the ocular pressure, improvement in blood-retinal barrier and visual guided behavioral functions after injection of GYY4137.

Another study employed a homozygous CBS knockout mouse model within their viable lifetime (CBS^−/−^, days 18–19) and compared H_2_S production in knockout and wild type animals [202]. Due to the total CBS knockout in the model, the relative contribution of CBS to the overall H_2_S production could be measured precisely. This study examined metabolites of the transsulfuration pathway, including cysteine and lanthionine, using MBB derivatization and LC-MS/MS. The results showed decreased H_2_S production down to 25% in the knockout liver extract compared to the WT liver tissue (0.2 U/mg of protein WT, 0.7 U/mg of protein KO). Findings from this study suggested that the majority of H_2_S formation in the liver is due to CBS and approximately 25% is maintained by CSE *and possibly 3*MST [202]. 

#### 4.2.2. Cystathione γ-Lyase (CSE)

Leigh et al. developed a murine model of CSE deficiency in order to elucidate the effect of H_2_S on in vivo renal erythropoietin (EPO) production [203]. They created a homozygous CSE^−/−^ knockout mice and compared them to wild type under different experimental conditions. First, the mice were subjected to either a 72 h period of hypoxia (11% O_2_) or normoxia (21% O_2_). During this period, the mice received two injections of either saline or the H_2_S-releasing molecule Na_2_S. Hypoxic wild type mice showed an increase in hemoglobin concentration (from 120 g/L to 180 g/L), in contrast to the hypoxic knockout mice. Similar hydrogen sulfide concentration was measured in hypoxic and normoxic mice after injections of either saline or Na_2_S. When comparing H_2_S concentrations of hypoxic and normoxic mice from the same mouse strain, no significant difference could be detected (0.15 nM WT normoxia, 0.16 nM WT hypoxia, 0.05 nM CSE^−/−^ normoxia, 0.06 nM CSE^−/−^ hypoxia). The injection of Na_2_S given to CSE^−/−^ mice in hypoxia significantly rescued hemoglobin concentrations and increased the concentration of circulating H_2_S [203]. On the other hand, under normoxic conditions, wild type mice exhibited significantly higher whole blood H_2_S concentration compared to CSE^−/−^ mice (0.15 nM WT, 0.05 nM CSE^−/−^). Normoxic CSE^−/−^ mice exhibited significantly higher EPO protein signal intensity than normoxic wild type mice (1.14 CSE^−/−^, 0.06 WT). During hypoxic conditions, this trend was reversed, and wild type mice exhibited a significantly higher signal intensity of EPO protein than their CSE^−/−^ counterparts (0.3 CSE^−/−^, 0.9 WT). In sum, H_2_S supplementation had a significant positive impact on EPO production during hypoxia but not during normoxia. In this way, Leigh et al. presented experimental evidence supporting the postulate that H_2_S regulates EPO production through the HIF-pathway [203]. H_2_S could therefore operate as an oxygen sensor during erythropoiesis.

In addition to the above described homozygous mouse model of CSE deficiency, Yang et al. generated heterozygous CSE^+/−^ and homozygous CSE^−/−^ mutant mice in order to investigate the role of H_2_S as a vasorelaxant and its impact on blood pressure [96]. An ion-selective electrode was used for determination of H_2_S concentrations. The concentration of H_2_S in aorta and heart tissue was reduced around 80% in CSE^−/−^ and around 50% in CSE^+/−^ mice. In serum, the H_2_S concentration was reduced around 50% in CSE^−/−^ and 20% in CSE^+/−^ mice [96]. Especially in the vascular system, CSE was reported as the main contributor to H_2_S formation [96]. The study revealed that mice devoid of CSE developed age-dependent hypertension, thereby providing evidence that H_2_S could be a regulator of blood pressure. 

#### 4.2.3. Cysteine Dioxygenase

The homeostasis of H_2_S was also investigated in a mouse model of *cysteine dioxygenase (CDO)* deficiency, an enzyme that has a major role in the catabolic pathway of cysteine [204]. CDO catalyzes the dioxygenation of cysteine to cysteine sulfinate in mammals. Cysteine can also undergo the previously described desulfhydration reaction, which is mainly catalyzed by the transsulfuration enzymes producing H_2_S as a side product. Indirect H_2_S markers such as thiosulfate, COX inhibition, hypotaurine and taurine were determined in CDO-deficient animals. Due to the complete lack of *CDO* activity, nearly all of the cysteine was metabolized by the transsulfuration enzymes, which resulted in several-fold elevated indirect H_2_S markers (e.g., thiosulfate: 420 µM/mg creatinine female KO, 320 µM/mg creatinine male KO, 100 µM/mg creatinine WT) [204].

#### 4.2.4. Rhodanese

Rhodanese is another component of H_2_S catabolism, which catalyzes the transfer of sulfane sulfur from glutathione persulfide (GSSH) to sulfite-generating thiosulfate, and from thiosulfate to cyanide-generating thiocyanate [134]. Rhodanese is abundant in kidney, liver and colon and it plays an important role in maintaining constant H_2_S concentration. Libiad et al. analyzed the specific polymorphic variants of *rhodanese* E102D and P285A expressed in murine liver concerning protein stability and kinetic activity [134]. The formation of H_2_S was measured in a turbidometric lead acetate assay followed by UV-spectrophotometry. Thiosulfate was quantified with the monobromobimane method and LC-MS/MS. For determination of thiosulfate-dependent H_2_S generation the authors used gas chromatography. Libiad et al. observed a more stable protein structure in the polymorphic variants compared to wild type [134]. The specific activity was several-fold higher in both variants (4- and 2.7-fold higher specific activities, thiosulfate-dependent H_2_S production: 1.2 µmol min^−1^g tissue^−1^) [134].
antioxidants-10-01065-t004_Table 4Table 4Determination of hydrogen sulfide production in genetic diseases of metabolism.MutationH_2_S FormationEnzyme ActivityBiological SampleMethodLiterature**Human Samples**




Pyridoxine non-responsive CBSDPlasma sulfide concentration was similar to the controlsAbsent activity of CBS→ Hcy elevation in plasma and urine, grossly decreased Cystat in plasma and urine, synthesis of H_2_S from grossly elevated Hcy by CSEPlasma samplesMonobromobimane(MBB) derivatization followed by HPLC separation and fluorescent quantitation[42]Thiosulfate was increased three timesUrineRMD due to cblG, cblE, cblJThiosulfate was increased 1.7 and 2.4 timesDecreased enzyme activity to 53% and 64% of the median in healthy controlsPlasma sample, urineMonobromobimane(MBB) derivatization followed by HPLC separation and fluorescent quantitation[42]ETHE1Plasma sulfide increased 5-fold with respect to reference rangeAbsent enzyme activityHuman plasmaMonobromobimane(MBB) derivatization followed by HPLC separation and HPLC-MS/MS measurements[27]T152I ETHE1 mutationMarked increase in H_2_S and thiosulfate concentration in both mutations3-fold lower activity of the enzyme, 4-fold decrease in Vmax, Km for GSSHunaffectedRecombinant human ETHE1 missing the N-terminal mitochondrial leader peptideMonobromobimane, reverse phase HPLC with a multi-signalfluorescence detector, mass spectroscopy[127]D196N ETHE1 mutation
2-fold higher Km for the substrate glutathione persulfide, around 15% decrease in Vmax, whereas 2-fold higher Km for GSSHETHE1^−/−^H_2_S concentration in all tissues much higher compared to WT: brain: 5-fold higher, muscle: 10-fold higher, liver: 10-fold higherthiosulfate in urine several-fold higher, sulfate lower, thiosulfate markedly increased in kidney, liver, muscle and brainTotal lack of ETHE1 activityEthe^−/−^ mice: liver, brain, kidney, external muscle, layers of colon, urineZinc acetate trap, gas chromatography/sulfur chemiluminescence detector[197,205]human (with ethylmalonic encephalopathy) blood and urineCBS p.P49L3-fold lower H_2_S formation in PLP-untreated CBS, similar formation by SAM stimulationStructural changes affect heme reactivity, enhanced affinity for CO, poor responsiveness to SAMHuman cultured fibroblastsAmperometry with H_2_S -selective electrode/coulometry (acetate method), UV-spectrophotometry[193,206]H_2_S formation similar to WT in PLP-treated CBS, impaired formation by SAM stimulation (1.1-fold vs. 1.9-fold in WT)CO inhibits the H_2_S producing activityVitamin B6 insufficiencyImpaired H_2_S production capacity because of the sensitivity of CSE to inactivation by loss of the PLP coenzymeLittle or no effect on CBS activity in the presence or absence of stimulation by SAM, CSE exhibited 70% lower activity, lower production of H_2_S-specific biomarkersHuman lysates of cultured hepatoma cellsGas chromatography with sulfur chemiluminescence detector[194]c.637G > A, a Glu213Lys variant of SQORc.446delTEpisodic accumulation of H_2_S, at high concentrationsH_2_S can coordinate to the iron in the heme a group of complex IV resulting in strong inhibitionCompared to WT, mutants exhibited: reduced SQOR protein and enzyme activity, unchanged sulfide generating enzyme concentration, reduced fibroblast SQOR enzyme activity and protein concentration, reduced SQOR protein and enzyme activity, unchanged sulfide generating enzyme concentration, decreased complex IV activity, but normal complex IV protein concentration and complex formationIsolated mitochondrial membrane fractions from human liver, lysates derived from peripheral blood mononuclear cells and from fibroblastsUV-spectrophotometry, monitoring of the decrease in absorbance at 278 nm due to reduction of CoQ1[131,195]**Animal models**




Cysteine dioxygenase knockout CDO^−/−^4.5-fold elevated concentration of urinary thiosulfate, lower abundances of COX4 and COX5b in liver, pancreas and kidney, slight elevation in plasma sulfate concentration, excess H_2_S/HS^−^ productionComplete lack of CDO activityMouse liver, kidney, brain, pancreasIndirect measurement of H_2_S (i.e., thiosulfate excretion, cytochrome c oxidase (COX) inhibition): taurine and hypotaurine were measured by HPLC, COX4 and COX5 by Western blotting[204]Homozygous (CSE^−/−^) and heterozygous (CSE^−/+^) mutant miceHomozygous mice compared to WT exhibited: endogenous H_2_S concentration in aorta and heart decreased by about 80%, serum H_2_S concentration by about 50% Heterozygous mice compared to WT exhibited: H_2_S concentration in aorta and heart decreased by about 50%, serum H_2_S concentration decreased by about 20%CBS^−/−^: Complete lack of enzyme activityCSE^−/+^: around 50% lower CBS activity versus wild typeMouse heart, aorta, vascular system,Ion-selective electrode on a Fisher Accumet Model 10 pH meter[96]CBS heterozygous knockout (CBS^−/+^) miceSignificantly reduced H_2_S concentration by 30% and 46% compared to wild type in male and female CBS^−/+^Around 50% CBS activity compared to WTMouse liverMethylene blue method, HPLC and quantification by fluorescence detector (measurement of thiol metabolites)[200]CBS treated with the Ad-lacZ virus (Ad-hCBS)Increased H_2_S concentration in plasma by 1.8-foldIncreased CBS activity by 1.4-fold, reduced homocysteine concentration significantly by 5.3-foldE102D variant of rhodaneseP285 A variant of rhodaneseH_2_S production in murine liver lysate is lowMore stable than wild type, exhibition of 4- and 2.7-fold higher specific activity, lower Kmfor cyanide for the E102D, but higher for the P285A, higher Km for thiosulfate for E102D, but lower for P285A, 17- and 1.6-fold higher catalytic efficiency (kcat/Km(CN)) in the cyanide detoxification assay, 1.6 and 4-fold lower sulfur transfer reaction from GSSH to sulfite, similar kcat/Km GSSH valuesMurine liver, human blood (wild type human rhodanese and recombinant variants)Turbidimetric lead acetate assay, UV-spectrophotometry, monobromobimane, reverse phase HPLC column, MS (detection of thiosulfate)thiosulfate-dependent H_2_S generation: gas chromatography[134]CSE^−/−^ mice on a C57BL/6 background (in hypoxia for a 72 h period)injection of either salineReduced H_2_S concentration in mutant compared to WTLower hemoglobin concentration, higher EPO concentration than wild type mice during normoxia, but lower during hypoxia, hypoxia downregulates CBS expressionMouse blood, kidney, human urinesulfide/H_2_S -sensitive microelectrode system (ion electrode)HPLC (thiosulfate concentration)[203]or Na_2_SComparable blood H_2_S concentration in mutants and wild typeRescue of hemoglobin concentration, rescued EPO concentration during hypoxia, increased CBS protein concentration in hypoxic CSE^−/−^CBS^+/−^ (B6.129P2-CBStm1Unc/J 002853)H_2_S added exogenously via GYY4137, a slow H_2_S donorDecreased CBS, CSE and MTHFR expressions, GYY4137 treatment rescues their concentration, GYY4137 reduces oxidative stress marker concentration and total GSH concentration, it normalizes overall glutamate concentration, it reduces ocular pressure back to the baseline levels and enables less vessel density and permeability in the eyeMouse blood, retinal tissueGYY4137 (crystalline solid in nature) dissolved in phosphate buffered saline (PBS) and injected intraperitoneally to mice[201]CBS-knockout (KO) mice~25% of H_2_S -forming activity when no activity in presence of CSE inhibitor PAGTotal lack of CBS activityMouse liver, kidney, brainMonkey plasma (WT)LC-MS/MS, HPLC[202]

## 5. Current Limitations and Future Studies

Hydrogen sulfide is a gasotransmitter with high volatility and a very short half-life due to its reactivity with low- and high-molecular weight biological targets. The chemical nature of H_2_S makes it difficult to determine its real concentration in biofluids, cells and tissues. Both in vitro and in vivo studies suggest that H_2_S concentrations vary quickly in response to stimuli, which demands a continuous, real-time measurement. Different methods for the determination of H_2_S exhibit widely different detection limits and sensitivities, partly due to interfering substances, which poses a challenge to its accurate quantitation. Aside from this caveat, not all methods capture all major species of H_2_S, that is, H_2_S/HS^−^. The absolute concentrations of H_2_S measured in different studies with different methods cannot therefore be compared side-by-side. Currently, monobromobimane derivatization is seen as the gold standard method, despite the previously mentioned disadvantages and the complicated sample pre-analytics required for quantification. Nevertheless, the summary presented in Table 3 enables a quick overview of currently available methods, which should assist investigators in deciding which setting fits best for their own research question. A yet to be developed method for the specific, reproducible, simple, sensitive and affordable determination of H_2_S in real time would sort out many of the currently unexplained variations in H_2_S steady-state concentrations and access the rate of H_2_S production in biological systems. Given the short-lived nature of H_2_S and its limited accumulation in specimens that are available for human research studies (plasma, urine, cells), methods for the direct, time-resolved measurement of H_2_S production may be more informative than end-point determinations of H_2_S concentration. 

## 6. Conclusions

Our knowledge of the biochemistry of hydrogen sulfide, the effects of exogenous addition of H_2_S and donor molecules to cultured cells, tissues and laboratory animals, as well as the development of methods for its detection, have deepened steadily over the last decades. Conversely, the importance of hydrogen sulfide biosynthesis and catabolism and its impact on physiology in vivo (patient samples, animal models of H_2_S disruption) have been examined by a few research groups only. Rare monogenic metabolic diseases that affect H_2_S biosynthesis and catabolism provide an excellent opportunity to investigate the roles of H_2_S in physiologically relevant compartments and organs, including the gut microbiome. For example, numerous mutations in the CBS gene and in other enzymes involved in sulfur amino acid metabolism have been described, yet only a handful of studies determined H_2_S concentrations in these patients. As a gasotransmitter involved in vascular dilation and cognitive functions, impairments in H_2_S homeostasis could contribute to the clinical manifestations of homocystinuria, cystathioninuria and other illnesses of sulfur metabolism. Further, knowledge of the specific rates of production, compartmentalization and reactivity of H_2_S that occur under different pathological conditions are essential for the design of new treatments and a better understanding of the respective diseases. 

## Figures and Tables

**Figure 1 antioxidants-10-01065-f001:**
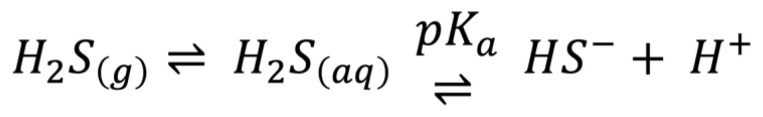
Ionization properties of H_2_S. Hydrogen sulfide exists as an equilibrium of its gaseous and aqueous forms. Deprotonation of H_2_S to form HS^−^ has a pK_a_ of 6.9–7.1.

**Figure 2 antioxidants-10-01065-f002:**
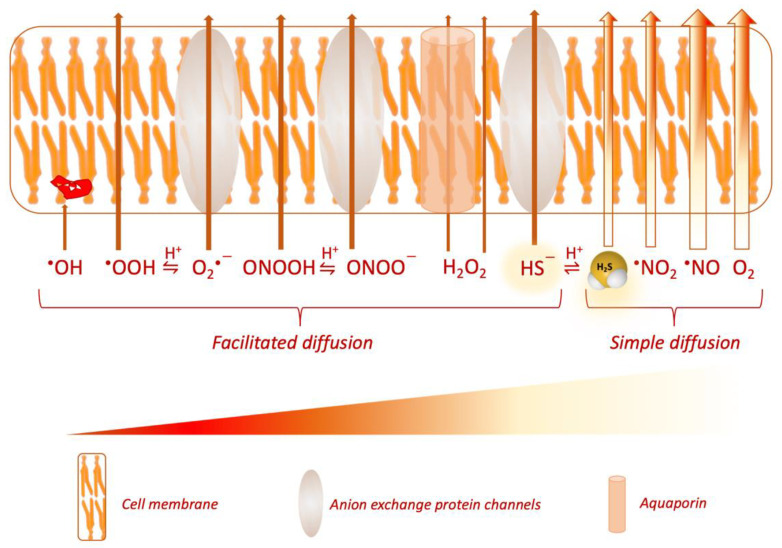
Permeability of H_2_S across biological membranes. The permeability of membranes to the different reactive species is proportional to their corresponding arrow size. Based on the studies and classification published by Möller et al. [29], H_2_S mobilizes across lipid membranes primarily by simple diffusion. Studies on erythrocyte membranes showed that mobilization of HS^−^ occurs by facilitated diffusion via anion exchange protein AE1 [31]. In contrast, oxidants such as ONOO^−^ and O_2_^•−^ require anion channels to traverse biological membranes. Mobilization of H_2_O_2_ across membranes is tightly controlled by cells. The reactivity of hydroxyl radical (^•^OH) outcompetes diffusion, which results in marked targeting of components of the membrane surface. Figure modified from Möller et al. [29].

**Figure 3 antioxidants-10-01065-f003:**
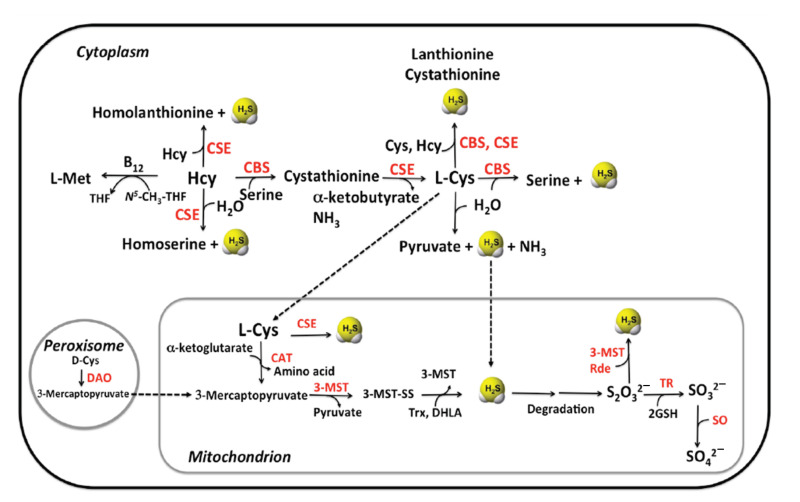
Biogenesis of H_2_S. Overview of reactions and compartments for the biosynthesis of H_2_S. CBS: Cystathionine β-synthase; CSE: Cystathionine γ-lyase; DAO: D-amino acid oxidase; 3MST: 3-mercaptopyruvate sulfurtransferase; CAT: Cysteine aminotransferase; Rde: rhodanese; SO: sulfite oxidase; Trx: thioredoxin. Modified from Libiad et al. [65].

**Figure 4 antioxidants-10-01065-f004:**
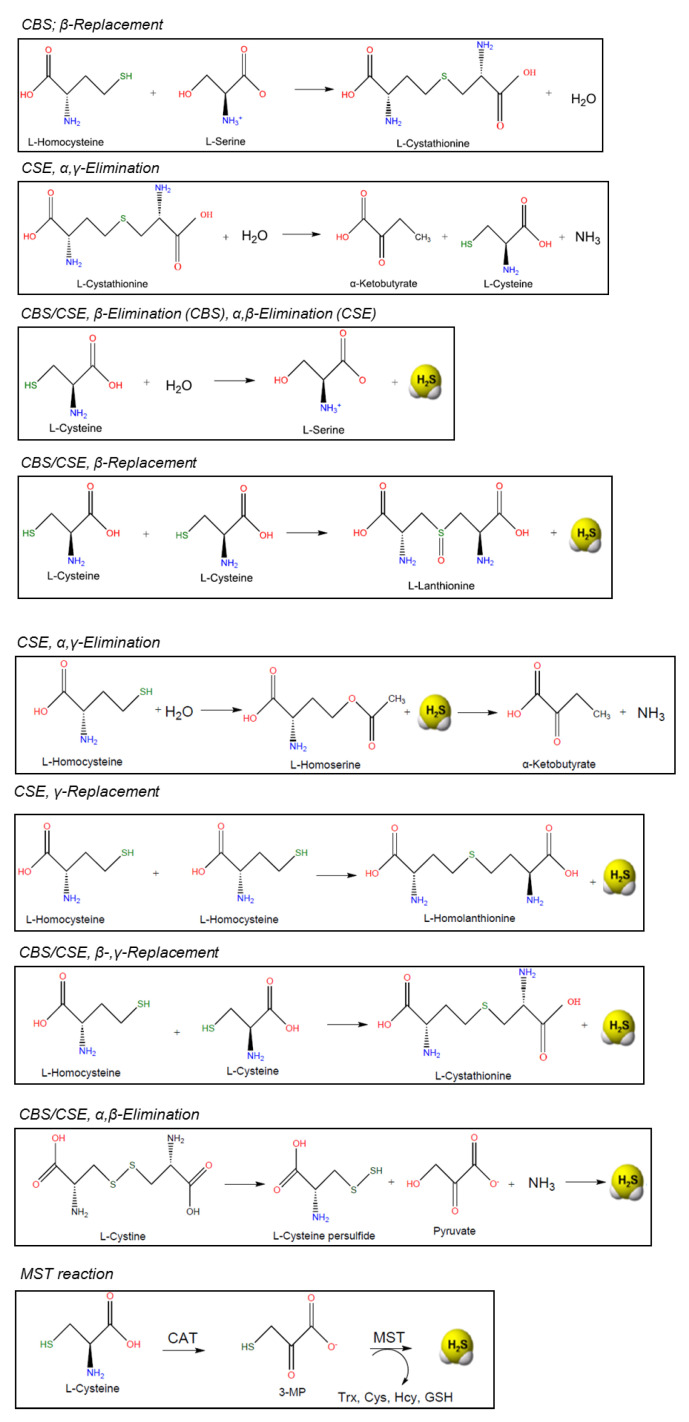
Canonical reactions of cystathionine-β-synthase (CBS), cystathionine-γ-lyase (CSE) and 3-mercaptopyruvate sulfurtransferase (MST). Modified from Giuffre and Vicente [10].

**Figure 5 antioxidants-10-01065-f005:**
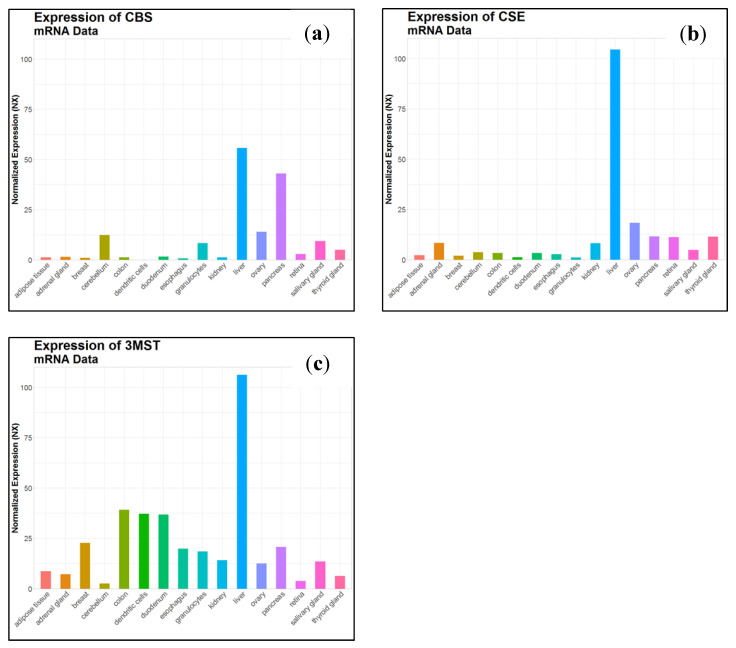
Normalized expression (NX) in different tissues of *CBS* (**a**), *CSE* (**b**), *3MST* (**c**) ([107]; source data for the plots were downloaded from the Human Protein Atlas, v19.proteinatlas.org).

**Figure 6 antioxidants-10-01065-f006:**
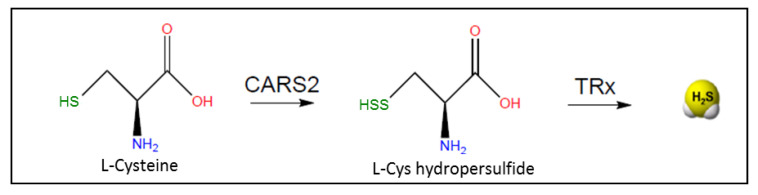
CARS2 reaction contributes to H_2_S formation in cells (CARS2 = *cysteinyl tRNA synthetase,* TRx *=* thioredoxin). Inspired by Akaike et al. [112].

**Figure 7 antioxidants-10-01065-f007:**
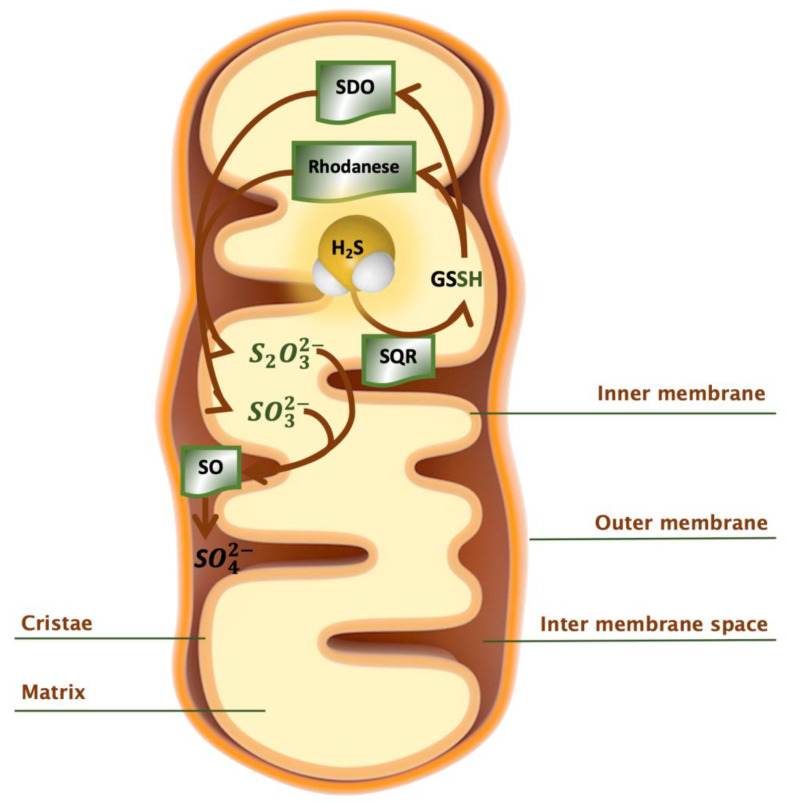
Oxidation pathway of H_2_S. In the mitochondrial matrix, SQR oxidizes H_2_S to GSSG and transfers the derived electrons to the respiratory chain, where they reduce coenzyme Q and drive ATP synthesis. GSSG undergoes further oxidation by SDO forming sulfite. Alternatively, rhodanese catalyzes GSSG, forming a thiosulfate anion. In the next step, the products sulfite and thiosulfate anions are oxidized to sulfate by the enzyme sulfite oxidase. (SQR = *sulfide quinone oxidoreductase,* SDO = *persulfide dioxygenase,* SO = *sulfite oxidase,* GSSH = glutathione persulfide). Modified from Libiad et al. [65].

**Figure 8 antioxidants-10-01065-f008:**
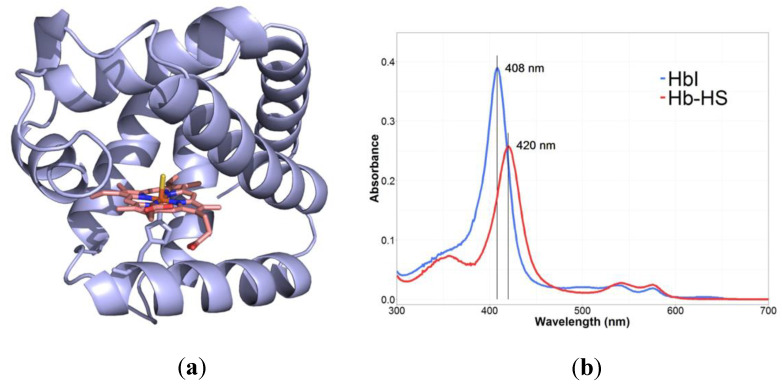
(**a**) Structure of HbI of *Lucina pectinata* bound to sulfide. (**b**) UV-visible spectra of hemoglobin I (10 µM) before (blue) and after (red) addition of NaHS (1 mM, source of H_2_S) in 0.05 M Tris buffer, pH = 8.6. Formation of the HbI-HS complex results in a red-shift in the UV-visible spectrum from 408 to 420 nm and hypochromy of the Soret band. Absorbance maxima are in agreement with data published in Boffi et al. [158].

**Figure 9 antioxidants-10-01065-f009:**
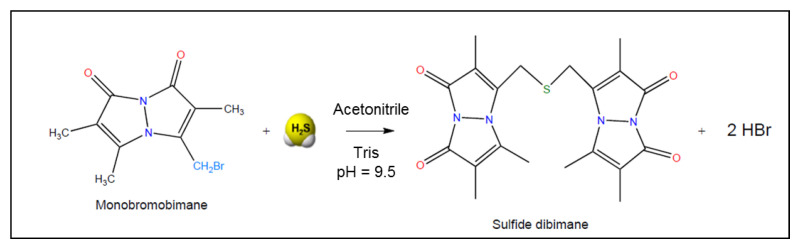
Monobromobimane derivatization reaction. Modified from Shen et al. [168].

**Table 1 antioxidants-10-01065-t001:** Concentration of H_2_S in biological compartments.

Specimen Type	[H_2_S]	Methods of Detection	Reference
**(1) Animal**			
Mouse brain	14 nM	Gas chromatography/chemiluminescence sulfur detector	[32]
Mouse liver	17 nM	Gas chromatography/chemiluminescence sulfur detector	[32]
Mouse plasma	0.7 μM	Monobromobimane assay	[33]
Mouse whole blood	15 nM	GC method and chemiluminescence detection	[34]
Rat whole blood	0.7 μM	Monobromobimane assay	[35]
Rat aorta	10 nM	Polarographic electrode PHSS	[36]
Rat liver and heart	>100 nM/g	Photometric detector and ion chromatography	[37]
Rat brain	9 µM	Gas chromatography/silver particles	[38]
Rat homogenate tissue	6 pM	Gas chromatography/silver particles	[38]
**(2) Human**			
Human breath	0.2 ng/L (0.1 ppb)	Chromatography	[39]
Human saliva	1.641–7.124 µM	Fluorescent probe	[40]
Human brain tissue	100 µM	Gas dialysis/ion chromatography	[41]
Human cerebrospinal fluid	3.48 ± 1.68 mg/L (102 ± 49 µM)	Selective electrochemical detection with ion chromatography	[22]
Human cerebrospinal fluid (pediatric patients)	8.71 ± 1.25 µM	Electrode (PXS-270, Shanghai Leici, China)	[21]
Human whole blood	100 pM	Gas chromatography/chemiluminescence sulfur detector	[32]
Human plasma (pediatric patients)	45.55 ± 6.55 µM	Electrode (PXS-270, Shanghai Leici, China)	[21]
Human plasma	300–500 nM	Monobromobimane (MMB) assay	[25]
	400–1000 nM	Monobromobimane assay	[42]
	70–125 μM	Microfluidic device with 2,6-dansyl azide (DNS-Az) probe	[20]
	c.a 5–100 µM	Methylene blue/UV-visible spectrophotometry	[17]
	54.0 ± 26.4 μM	Methylene blue/UV-visible spectrophotometry	[18]
	14.11 ± 5.24 μM	Methylene blue/UV-visible spectrophotometry	[19]

**Table 2 antioxidants-10-01065-t002:** Reported effects of H_2_S in selected cell types.

Cell Type	H_2_S Dosage	Mechanism	Response	Reference
Neurons, glia	Supraphysiological	Stimulation of NMDA receptors, induced by an increase of cAMP concentration	Causes/contributes to tissue damage, neuroinflammatory	[43]
	Physiological		Messenger in response to specific stimuli (usually noxious)	[43]
	Physiological	Increasing the concentration of glutathione, activating K^+^ATP and Cl^−^ channels	Protection from oxidative stress	[46,48]
Vascular endothelial cells	Release of H_2_S	Membrane hyperpolarization (opening of the VSMC K^+^ATP channels), reduction of extracellular Ca^2+^ entry	Vasorelaxant effect	[49]
	Exogenous H_2_S	Suppression of VSMC proliferation through protein kinase (MAPK) signaling pathway	Attenuation of cell proliferation rate, increase of cell apoptosis, regulation of cell growth, structural remodeling of vessel tissues	[49,50]
	Reduced (e.g., due to CSE deficiency)	Loss of regulatory effects	Hypertension, atherosclerosis, myocardial injury	[51]
Pulmonary artery cells	Endogenous release	Inhibition of proliferation and interleukin- (IL)-8 secretion by impairing extracellular signal-regulated kinase 1/2 (ERK1/2) and p38-dependent signaling pathways	Lung remodeling, inhibiting collagen accumulation in the wall of the pulmonary artery in hypoxia	[52]
	Inhibition of endogenous H_2_S release	Signal-regulated kinase 1/2 (ERK1/2) and p38-dependent signaling pathways active	Increased human airway, smooth muscle proliferation, IL-8 secretion	[53]
Pancreatic islets cells	Physiological	Stimulation of K^+^ATP channels in β-cells, possible inhibition of insulin release by activation of K^+^ATP channels	Inhibition of insulin release, induction of apoptosis	[54,55]
Colonocytes	Release of H_2_S	Activation of K^+^ATP channels in the vascular tissue, stimulation of vanilloid type I receptor	Inhibition of jejunum and colon motility, regulation of the gastric mucosal blood flow, secretion and increased spike activity in the afferent neurons of the enteric nervous system	[56,57]
	Supraphysiological	Inhibits cytochrome c oxidase and synthesis of ATP	Pro-inflammatory, suppresses mitochondrial energy production, destabilizes the protective intestinal mucus layer, potentially participates in the etiology of bowel disorders, occurs as genotoxic agent in colorectal cancer	[14,44,58]
	Low concentration	Energy substrate	Anti-inflammatory	[45]

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
