# Peer review of "Biosynthesis, Quantification and Genetic Diseases of the Smallest Signaling Thiol Metabolite: Hydrogen Sulfide"

_antioxidants, 2021, doi:10.3390/antiox10071065_

Round 1
Reviewer 1 Report
1.) This is a comprehensive review. However, according to Antioxidants scope: “Antioxidants provides an advanced forum for studies related to the science and technology of antioxidants.”
The manuscript mentions the word antioxidant only once (lines 881-883).
Therefore, it would be appropriate to include in the revised manuscript information on the involvement of H2S signaling, biogenesis, CBS, CSE and 3MST enzyme activities and the microbiome in the antioxidant biological system.
It would be useful to mention this also in “5. Current limitations and future studies” (line 955).
2.) The manuscript cites 177 works, of which only 4 from 2020 and none from 2021.
Therefore, it would be appropriate to update the list of citations of the relevant works from 2020 and 2021 and to include the new results in the revised manuscript.
3.) It is emphasized that: “disease-promoting effects of H2S largely depend on its concentration and compartmentalization” (line 15) or “concentration of H2S in biological compartments are in a wide range in Table 1” (line 97).
Therefore, more details concerning the dynamic H2S concentrations in “micro space” (e.g. in several nm3) could be included if available.
Author Response
Reviewer 1
1.) This is a comprehensive review. However, according to Antioxidants scope: “Antioxidants provides an advanced forum for studies related to the science and technology of antioxidants.”
The manuscript mentions the word antioxidant only once (lines 881-883).
Therefore, it would be appropriate to include in the revised manuscript information on the involvement of H2S signaling, biogenesis, CBS, CSE and 3MST enzyme activities and the microbiome in the antioxidant biological system.
It would be useful to mention this also in “5. Current limitations and future studies” (line 955).
Response: We agree with this important point concerning scope of the journal and also with the genuine need to incorporate the involvement of H2S in the antioxidant systems to enrich the content of the work. To address this with clarity, we have now incorporated a new section entitled “2.6. Antioxidant functions of H2S“, which covers major routes for antioxidant activity of H2S and cites original manuscripts and reviews on the topic.
2.) The manuscript cites 177 works, of which only 4 from 2020 and none from 2021.
Therefore, it would be appropriate to update the list of citations of the relevant works from 2020 and 2021 and to include the new results in the revised manuscript.
Response: We apologize for this oversight that occurred upon our failure to check for new references post-bulk of the writing, which took place during the first half of 2020. We have now reviewed the literature to ensure that new works published in 2020 and 2021 are described and cited. The revised version of the review now contains a total of 205 references, including 11 from 2020 and 9 from 2021.
3.) It is emphasized that: “disease-promoting effects of H2S largely depend on its concentration and compartmentalization” (line 15) or “concentration of H2S in biological compartments are in a wide range in Table 1” (line 97).
Therefore, more details concerning the dynamic H2S concentrations in “micro space” (e.g. in several nm3) could be included if available.
Response: Thanks for pointing that out. We have now supplemented our discussion on compartmentalization. The description of the 3D model of the mammalian cell presented in line 97 has been expanded and additional findings are now presented. This section now includes: (i) further details on passive and facilitated diffusion, (ii) the discrepancy between reported concentrations of H2S in plasma and what is actually feasible from mathematical predictions and the known chemical properties of H2S/HS-, and (iii) the primary role of local metabolization in controlling H2S concentrations over what can be achieved via diffusion to other compartments.

Reviewer 2 Report
General Comments:
This is a grammatically well-written review H2S, although it is mostly a repeat of many other (and more detailed) reviews and most sections offer little new insight. Section 4 is an exception and it essentially rescues the ms.
Minor comments:
Page 2, line 52: The comment that 70-80% H2S exists as HS- (anion) is only correct for plasma where pH is 7.4. Intracellular (general cytosol) pH is 6.9-70. So this is reduced to 50%.
Page 2, line 79: Agreed they have decreased over time but there are still numerous reports within the past year that claim plasma [H2S] > 10 uM. These need a strong comment against.
Page 2, line 88: But see Olson Respir. Physiol. Neurobiol. 186:173, 2013.
Figure 2, re H2S diffusion only; see Jennings Am. J. Physiol. Cell. Physiol. 305:C941, 2013
Page 6: This first paragraph tends to wander. Not clear what the point is.
Page 13, Figure 6 : I’m not sure the SSH in this figure is correct, please check.
Page 13, line 346: As far as I know reference 106 never measured H2S or reported on [H2S]. Also see Linden and others. The level of SQOR is very high here to prevent H2S from entering the blood.
Page 15, line 414: Can this really be a source of H2S at this pH? Please comment.
Page 21: It would be helpful to give the common acronyms for these fluorophores (as was done for AzMC).
Author Response
Reviewer 2
General Comments:
This is a grammatically well-written review H2S, although it is mostly a repeat of many other (and more detailed) reviews and most sections offer little new insight. Section 4 is an exception and it essentially rescues the ms.
Response: Thank you. We felt that the overlap with content of existing reviews is a necessary compromise to provide context. We appreciate that the reviewer valued section 4 as a novel revision in the field. In addition to section 4, we briefly discussed the formation of small oxoacids of sulfur (section 1), which to our knowledge, has not been covered elsewhere. We also incorporate a report based on Raman studies that failed to find evidence for the existence of S2- in aqueous media (section 1). We hope that readers would find our review of the literature a useful tool to more quickly find existing research and in depth-reviews by colleagues.
Minor comments:
Page 2, line 52: The comment that 70-80% H2S exists as HS- (anion) is only correct for plasma where pH is 7.4. Intracellular (general cytosol) pH is 6.9-70. So this is reduced to 50%.
Response: Thank you, we corrected this as suggested.
Page 2, line 79: Agreed they have decreased over time but there are still numerous reports within the past year that claim plasma [H2S] > 10 uM. These need a strong comment against.
Response: We agree. We included recent works still reporting high concentrations of plasma H2S (please, see updated values of human plasma H2S, as well as new values of H2S in human cerebrospinal fluid, Table 1), and have now discussed this problematic overestimation of H2S more critically. We also agree with the reviewer that readouts above 10 µM free H2S in plasma are an overestimation. The steady state concentration of H2S in plasma, if it exists, is probably closer to pM to nM.
Page 2, line 88: But see Olson Respir. Physiol. Neurobiol. 186:173, 2013.
Response: Thanks for pointing us to this important work. We have added a paragraph right after the discussion of H2S as a paracrine signal, to point readers to a matter that remains an experimental challenge. Section “1.2. Concentration, signaling and cellular response to H2S” has been edited substantially to compare and contrast experimental findings and predictions of H2S diffusion and compartmentalization. We appreciate the reviewer helping us to enrich this section!
Figure 2, re H2S diffusion only; see Jennings Am. J. Physiol. Cell. Physiol. 305:C941, 2013
Response: Thank you for pointing to our accidental omission of this work. We updated the discussion on H2S and HS- diffusion citing the work of Jennings also within section 1.2, and modified Figure 2 accordingly to depict these transport mechanisms. This section now reads as follows:
“In vitro studies indicated that both H2S (simple diffusion) and HS— (facilitated diffusion by anion exchange protein AE1 in erythrocytes) produced by tissues enter red blood cells rapidly, thereby acting as a sink for H2S and lowering extracellular concentration of the gasotransmitter [31]. As can be appreciated from these in-depth experimental studies and predictions, the properties and relationships of the sources and sinks of H2S are complex. The extent and effects of H2S mobilization in the biological milieu require further in-vestigation”.
Page 6: This first paragraph tends to wander. Not clear what the point is.
Response: Thanks for noting this. The following text disrupted reading flow in that section and was deleted for clarity (as this is cited in page 2):
“The reactivity and synergy of H2S with other gasotransmitters such as NO and CO have also been described and carefully reviewed [10]. Results from studies by Mishanina et al, indicated that the pathway of H2S oxidation could mediate sulfide signaling, and that the transsulfuration enzymes, which are major sources of H2S, ensure target specificity [46].”
Page 13, Figure 6 : I’m not sure the SSH in this figure is correct, please check.
Response: Thanks for noting this error on Figure 6! This was an accidental copy-paste duplicate of the substrate. This has now been corrected to show formation of product L-Cys hydropersulfide (SSH) from substrate L-Cys (SH).
Page 13, line 346: As far as I know reference 106 never measured H2S or reported on [H2S]. Also see Linden and others. The level of SQOR is very high here to prevent H2S from entering the blood.
Response: That was a citation error. We meant to cite reference 107, which covers the studies with regularmicrobiome and germ-free mice. We corrected this, as well as the duplication of references 107 and 108. Thank you also, for pointing us to the work by Linden et al. We incorporated those important findings in this section, as follows:
“Findings by Linden et al. support the proposal that high expression of SQOR in the colonic mucosa ensures rapid oxidation of H2S, which would substantially limit or pre-clude gasotransmitter transfer into the bloodstream [125]. The complexity of host-microbiome processing of H2S has been reviewed in detail [124] and calls for further investigation.”
Page 15, line 414: Can this really be a source of H2S at this pH? Please comment.
Response: Thank you, this section needed clarification. We added a new sentence, so this section now reads as follows:
“The critical pH below which H2S is released from acid-labile sulfur is 5.4 [23]. Thus, this reaction is unavailable as a source of H2S inside the mitochondrion, where pH > 7.8.”
Page 21: It would be helpful to give the common acronyms for these fluorophores (as was done for AzMC).
Response: Thank you for noting this. We incorporated common names and acronyms for the fluorophores whenever they were available.
Round 2
Reviewer 2 Report
I appreciate the author's efforts in responding to my comments.
One minor thing, I suggest modification (downward) of the enthusiasm for the monobromobimane assay. Please see the paper by Bogdandi et al. 2019 Br. J. Pharmacol. 176:646-670. This method also artificially inflates plasma H2S measurements.
Author Response
Thank you! We agree in full. We have taken the opportunity to bring this up explicitly. In addition to the reference by Bogdani et al, we included also references 176 and 177, in the following new paragraph in page 22:
“While the MBB method seems like an accurate, quantitative, and scalable measurement of discrete pools of hydrogen sulfide from primary volatile sulfide pools, it is not free of limitations. Apart from the multiple experimental steps required to measure the con-centration of H2S both in solution and in its gaseous phase, carefully controlled studies of reactivity revealed that MBB itself modifies sulfur speciation in biological specimens such as plasma and serum [59,176]. The concentration of sulfide determined by the MBB method in serum samples was found to depend on incubation time and on the concen-tration of MBB itself [59]. These findings are in line with an independent report that compared sulfur quantitation side-by-side using MBB, iodoacetamide (IAM), and N-ethylmaleimide (NEM) [177]. The readout of sulfide concentration in whole blood was higher with probes NEM and MBB compared to the same measurement utilizing IAM [177]. These differences can be explained by a greater capacity of NEM and MBB in cleaving polysulfur chains compared to IAM. These limitations have lead experts in the field to recommend “extraordinary care” with the reporting of “free sulfide levels” and with the assignment of roles to specific sulfur species in biological systems [59].“
- Bogdándi, V.; Ida, T.; Sutton, T.R.; Bianco, C.; Ditrói, T.; Koster, G.; Henthorn, H.A.; Minnion, M.; Toscano, J.P.; van der Vliet, A.; et al. Speciation of reactive sulfur species and their reactions with alkylating agents: do we have any clue about what is present inside the cell? Br. J. Pharmacol. 2019, 176, 646–670, doi:10.1111/bph.14394.
- Nagy, P.; Pálinkás, Z.; Nagy, A.; Budai, B.; Tóth, I.; Vasas, A. Chemical aspects of hydrogen sulfide measurements in physiological samples. Biochim. Biophys. Acta - Gen. Subj. 2014, 1840, 876–891, doi:10.1016/j.bbagen.2013.05.037.
- Sutton, T.R.; Minnion, M.; Barbarino, F.; Koster, G.; Fernandez, B.O.; Cumpstey, A.F.; Wischmann, P.; Madhani, M.; Frenneaux, M.P.; Postle, A.D.; et al. A robust and versatile mass spectrometry platform for comprehensive assessment of the thiol redox metabolome. Redox Biol. 2018, 16, 359–380, doi:10.1016/j.redox.2018.02.012.
Thank you for your time and effort in so carefully reviewing this work! We have added a statement in "Acknowledgements" to express our gratitude for the constructive feedback we received during the preparation of the revisions.